# Learning Reward Functions for Cooperative Resilience in Multi-Agent Systems

## Abstract

Multi-agent systems often operate in dynamic and uncertain environments, where agents must not only pursue individual goals but also safeguard collective functionality. This challenge is especially acute in mixed-motive multi-agent systems. This work focuses on **cooperative resilience**—the ability of agents to anticipate, resist, recover, and transform in the face of disruptions—a critical yet underexplored property in Multi-Agent Reinforcement Learning. We study how reward function design influences resilience in mixed-motive settings and introduce a novel framework that learns reward functions from ranked trajectories, guided by a cooperative resilience metric. Agents are trained in a suite of social dilemma environments using three reward strategies: (i) traditional individual reward; (ii) resilience-inferred reward; and (iii) hybrid that balance both. We explore three reward parameterizations—linear models, hand-crafted features, and neural networks—and employ two preference-based learning algorithms to infer rewards from behavioral rankings. Our results demonstrate that *hybrid strategy* significantly improve robustness under disruptions without degrading task performance and reduce catastrophic outcomes like resource overuse. These findings underscore the importance of reward design in fostering resilient cooperation, and represent a step toward developing robust multi-agent systems capable of sustaining cooperation in uncertain environments.

## 1 Introduction

Multi-agent systems often operate in dynamic and uncertain environments, where disruptions threaten overall functionality Topolewicz et al. (2023); Yildirim et al. (2019). In particular, in mixed-motive multi-agent systems, agents must do more than simply optimize individual performance, they must collectively adapt and recover from disruptions to preserve system-level well-being. Disruptions, whether internal (e.g., system failures), external (e.g., environmental shocks), or adversarial (e.g., targeted attacks), can compromise system performance, underscoring the need for adaptive recovery mechanisms Topolewicz et al. (2023). This motivates recent studies of resilience in multi-agent systems Chacon-Chamorro et al. (2025); Shraga et al. (2025), in particular the concept of **cooperative resilience**, defined as the ability of agents to sustain system-level well-being by anticipating, resisting, recovering, and transforming under disruption Chacon-Chamorro et al. (2025). Unlike traditional notions of stability, equilibrium, or robustness, cooperative resilience captures the dynamic, temporal, and distributed nature of multi-agent systems operating in uncertain and failure-prone environments.

Cooperative resilience represents a critical yet underexplored dimension in Multi-Agent Reinforcement Learning (MARL). Conventional MARL methods, including value factorization approaches such as QMIX Rashid et al. (2020) and policy gradient methods such as PPO Schulman et al. (2017), rely on the specification of a coherent reward function. This reward must capture the interdependent nature of the joint task shaped by agents' interactions. However, how the reward design influences the ability of agents to cooperate, adapt, and persist under adverse conditions is still not sufficiently understood. This limitation becomes particularly evident in mixed-motive settings, where agents must balance individual goals with collective outcomes, making the design of effective incentive mechanisms especially challenging Nie et al. (2024); Leibo et al. (2017).

In this work, we address this challenge using Inverse Reinforcement Learning (IRL) Goktas et al. (2025); Ashwood et al. (2022); Wu et al. (2022), a principled framework for inferring reward functions from observed agent behavior. IRL recovers latent reward functions that are assumed to have generated trajectories exhibiting desirable or near-optimal responses to disruptions. This approach enables the discovery of incentive structures that support emergent properties such as cooperative resilience, without requiring explicit encoding of such behaviors in the reward design. This perspective is complementary to recent work on group resilience through collaboration protocols Shraga et al. (2025), which seeks to improve resilience by designing mechanisms and interaction structures that make agents more robust to perturbations. In contrast, our goal is to *infer* a reward function that captures cooperative resilience directly from behavioral evidence. Protocol-based approaches enrich agents' capabilities, while resilience-aligned reward inference uncovers the underlying incentive structure that can support resilient collective behavior.

Building on this idea, we introduce an approach for learning reward functions from ranked agent trajectories scored with a cooperative resilience metric. This metric quantitatively evaluates how well trajectories preserve collective welfare in the presence of disruptions. By using it, we generate preference rankings over observed behaviors and feed them into a preference-based IRL pipeline to infer reward functions that implicitly encode cooperative resilience. In this way, our main contribution is a reward function design method that leverages cooperative resilience to infer a collective reward component, steering agents toward sustained system performance under disruptions. The learned reward is, in principle, compatible with different MARL algorithms, since it can be used as a drop-in replacement for the underlying reward signal. In this sense, our framework offers a flexible reward-design component that can complement existing methods.

We validate our approach in a mixed-motive social dilemma inspired by the Commons Harvest scenario from Melting Pot Perolat et al. (2017); Agapiou et al. (2022). The environment captures the tension between individual incentives to maximize resource consumption and the collective need to preserve shared resources. Our results show that resilience-inferred rewards foster adaptive behaviors under disruption, extending sustainability and improving collective outcomes compared to baselines PPO and QMIX. To assess scalability, we extended the evaluation to larger multi-agent environments with more agents and resource limitations, where our method enhanced cooperative resilience and, beyond that, improved overall system behavior by extending sustainability and collective well-being over time, without sacrificing individual task performance. This highlights our broader insight: reward functions can be treated as a form of *a priori* knowledge, extracted from trajectory analysis under a system-level cooperative resilience metric.

The remainder of this paper is organized as follows. Section 2 reviews related work, Section 3 presents our framework and Section 4 details the experimental setup and key results. Section 5 concludes with a summary and future directions. The appendix provides implementation details, extended results, and reproducibility information for experiments.

## 2 BACKGROUND AND RELATED WORK

### 2.1 COOPERATIVE AI AND RESILIENCE

Cooperative AI studies the design of multi-agent systems that achieve outcomes benefiting the group as a whole Dafoe et al. (2020); Hammond et al. (2025). In mixed-motive environments, where agents must balance individual objectives with collective welfare, designing mechanisms that foster cooperation is particularly challenging Hammond et al. (2025). Standard reinforcement learning approaches often emphasize individual performance, which can lead to selfish behavior and the degradation of shared resources, especially in the presence of social dilemmas Rios et al. (2023); Strümke et al. (2022); Du (2022); Leibo et al. (2017).

The introduction of disruptions, such as resource scarcity, unsustainable behaviors, or abrupt environmental changes, further complicates cooperation Jasper (2004); Orner et al. (2025). In these contexts, resilience becomes essential to sustain joint welfare. We focus on **cooperative resilience** Chacon-Chamorro et al. (2025), similar to the concept of group resilience Shraga et al. (2025), a system-level property that quantifies how well a group of agents can maintain collective well-being under stress. Building on approaches from ecology, infrastructure, and economic networks Ayyub (2014); Cimellaro et al. (2016); Gerges et al. (2022), the methodology in Chacon-Chamorro et al. (2025) evaluates resilience by comparing disrupted vs. baseline performance across indicators of

collective well-being, producing a score of this property. This score provides a quantitative basis for comparing systems and analyzing the behaviors and incentives that lead to resilient outcomes. It can also inform the design of new agents and reward structures that promote both cooperation and robustness.

## 2.2 MULTI-AGENT REINFORCEMENT LEARNING

Traditional Reinforcement Learning (RL) optimizes single-agent behavior through trial-and-error, but in multi-agent settings strategic interdependence makes learning more complex, giving rise to coordination, competition, and equilibrium. To address these challenges, MARL algorithms such as QMIX Rashid et al. (2020) and COMA Foerster (2018) tackle credit assignment, while social reward shaping methods Jaques et al. (2019); Hughes et al. (2018) encourage cooperation and mitigate free-riding by rewarding agents for influencing others' actions Jaques et al. (2019). More recent incentive-exchange mechanisms, such as peer reward Lupu & Precup (2020); Yang et al. (2020), norm formation through sanctions Vinitsky et al. (2023), and mutual acknowledgment protocols Phan et al. (2024), enable agents to deliberately influence the rewards of others, thus promoting pro-social behavior through structured interactions. These methods operate by designing incentives or communication protocols that induce cooperation.

However, all of these approaches presuppose the availability of a coherent reward structure. In mixed-motive environments, this becomes particularly challenging: the reward must balance individual and collective welfare, and under disruptive conditions the relevant system-level property is *cooperative resilience*. Designing such reward functions directly is difficult. This motivates the use of inverse reinforcement learning (IRL), where rewards are *inferred* from observed behaviors rather than engineered.

## 2.3 INVERSE REINFORCEMENT LEARNING

Inverse Reinforcement Learning (IRL) is a framework for infering reward functions from behavior Adams et al. (2022); Metelli et al. (2023); Arora & Doshi (2021). A major limitation is its reliance on demonstrations being optimal, which is rarely true in practice Brown et al. (2019); Goktas et al. (2025); Poiani et al. (2024). Extending IRL to multi-agent settings (MAIRL) introduces further challenges such as joint action spaces and equilibrium-based formulations Natarajan et al. (2010); Littman (1994); Çelikok et al. (2024), which become intractable in complex or disrupted domains Goktas et al. (2025). Alternatives like swarMDP Šošić et al. (2017) learn local rewards but struggle to generalize. To address this, we adopt preference-based IRL Brown et al. (2019); Willis et al. (2025), which learns from trajectory comparisons and is well-suited to cooperative resilience, where behaviors can be naturally ranked.

## 3 PROBLEM FORMULATION AND METHODOLOGY

We consider multi-agent environments where agents interact under mixed-motive conditions and share access to common resources. The environment is modeled using the standard *joint-state, joint-action* formulation of a Markov game (or multi-agent MDP), defined by the tuple $(S, A, P, R, \gamma)$, where $S$ is the global environment state, $A = A_1 \times \cdots \times A_n$ is the joint action space, $P$ is the transition function over the joint state–action space, $R$ denotes the reward structure, and $\gamma$ is the discount factor. This formalization is consistent with the multi-agent decision-process models presented in Littman (1994); Boutilier (1996). Each agent executes its own decentralized policy $\pi_i(a_i \mid s)$.

In this setting, standard reward functions $R$ often prioritize short-term individual gains, which can undermine collective welfare, especially under disruptive conditions. This motivates our central objective: to learn a reward function that promotes cooperative resilience. To this end, we propose a two-step methodology: (i) ranking trajectories using a cooperative resilience metric (see Subsection 3.1), and (ii) learning a reward function from preferences using one of two methods—margin-based optimization or probabilistic modeling (see Subsection 3.2).

Figure 1 summarizes the proposed learning pipeline. The process begins with a system of interacting agents operating in a mixed-motive environment, as illustrated in panel (a). In this example setup, agents harvest resources from a shared apple tree in a grid-world, balancing individual consumption

with long-term sustainability. Panel (b) presents the reward learning pipeline. First, agent trajectories are collected and evaluated using a cooperative resilience metric. This evaluation induces a ranking over trajectories based on their resilience scores. Next, this ranking is used as input to a preference-based IRL module that learns a reward function aligned with resilient behaviors. The learned reward is then integrated into the agents' policy learning process, guiding behavior in future interactions with the environment.

### 3.1 RANKING TRAJECTORIES BY COOPERATIVE RESILIENCE

A trajectory $\tau = (s_0, a_0, s_1, a_1, \cdots, s_T)$ is defined as a sequence of states and joint actions over a time horizon $T$, where $s_t \in S$ and $a_t \in A$ denote the state and joint action space at time step $t$, respectively. Let $\mathcal{D}$ be the set of trajectories generated by agents interacting with the environment under a given policy. We adopt the methodology from Chacon-Chamorro et al. (2025) to compute a resilience score $\rho(\tau)$ for each trajectory $\tau \in \mathcal{D}$. A score of $\rho = 1$ indicates alignment with a baseline without disruption, values below 1 reflect loss of resilience, and scores above 1 suggest exceptional recovery or improved performance after disruption.

To rank these trajectories, we assign a resilience score $\rho(\tau)$ to each $\tau$. This score is based on a set of performance curves derived from four types of collective well-being indicators: **Cumulative consumption**, computed *per agent* (yielding one curve per agent); **Resource availability**, measured as the total number of apples present in the environment; **Gini index** of the consumption distribution, capturing inequality; and a **Hunger index**, estimating the delay between successive accesses to resources. The indicators we employ should be viewed as an instantiation of a more general template: the framework remains applicable as long as practitioners define the dimensions of collective well-being that matter for their particular domain.

Each indicator is measured under two conditions: a *baseline* scenario with no disruption, and a *disrupted* scenario. Let $I_k(t)$ denote the indicator $k$ over time, $t_d$ be the disruption time, $t_f$ the time of worst degradation, and $t_r$ the recovery endpoint. In practice, $t_d$ is common to all indicators, while $t_f$ and $t_r$ are computed independently for each indicator to capture their distinct degradation and recovery dynamics. Consistently with Ayyub (2014) formulation, we set $t_f = \arg\min_{t \geq t_d} I_k^{\text{disrupted}}(t)$ within the considered window, and take $t_r$ as either the end of the horizon (single disruption) or the last timestep before the next disruption (multiple disruptions).

We define the failure and recovery profiles as: $\text{FailureProfile}_k = \int_{t_d}^{t_f} I_k^{\text{disrupted}}(t)(I_k^{\text{baseline}}(t))^{-1}dt$, $\text{RecoveryProfile}_k = \int_{t_f}^{t_r} I_k^{\text{disrupted}}(t)(I_k^{\text{baseline}}(t))^{-1}dt$, and denote the durations $\Delta t_f = t_f - t_d$, $\Delta t_r = t_r - t_f$. Using these quantities, the resilience score for indicator $k$ is computed as:

$$\rho_k = \frac{t_d + \text{FailureProfile}_k \cdot \Delta t_f + \text{RecoveryProfile}_k \cdot \Delta t_r}{t_d + \Delta t_f + \Delta t_r}.$$

Finally, we aggregate the indicator-specific resilience scores into a global resilience score using the harmonic mean $\rho(\tau) = \left(\frac{1}{K}\sum_{k=1}^{K}\frac{1}{\rho_k}\right)^{-1}$, which balances contributions across indicators by

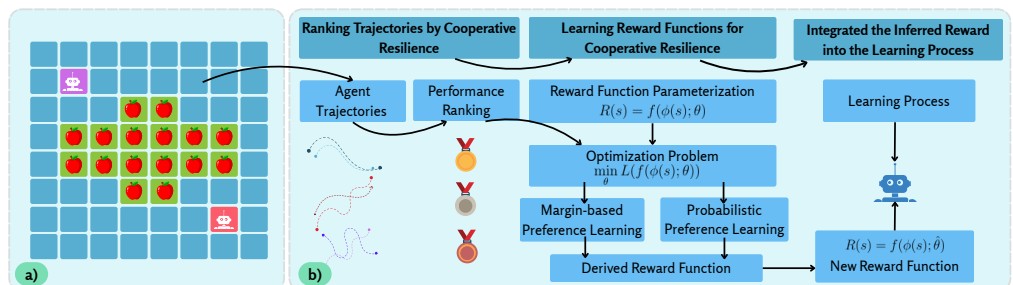

Figure 1: **(a)** Mixed-motive environment used throughout this study. Two agents interact in an $8 \times 8$ grid with a central apple tree containing 16 apples. **(b)** Overview of our proposed reward learning pipeline. This figure illustrates the full loop from data collection to policy learning.

penalizing low values, thus ensuring that system resilience is not dominated by a single dimension. Ultimately, this methodology captures cooperative resilience by contrasting system behavior with and without disruption, deriving failure and recovery profiles for each indicator, and aggregating them through the harmonic mean Chacon-Chamorro et al. (2025).

Once the scores are computed, we induce a preference ordering over the trajectories: a trajectory $\tau_i$ is preferred over $\tau_j$, if it exhibits a higher cooperative resilience score. Formally, the preference relationship can be expressed as $\tau_i \succ \tau_j$ if only if $\rho(\tau_i) > \rho(\tau_j)$, where $\rho(\tau)$ denotes the cooperative resilience score of the trajectory $\tau$. This preference structure provides the basis for learning reward functions aligned with cooperative resilience, as discussed in Subsection 3.2.

## 3.2 Learning Reward Functions for Cooperative Resilience

The next step is to learn a reward function $\hat{R} : S \to \mathbb{R}$ that aligns agent behavior with cooperative resilience, as indicated by the trajectory rankings. A crucial design decision at this stage is the parameterization of reward function. We assume that the reward function depends on a feature representation of the state, denoted as $\phi(s) : S \to \mathbb{R}^n$, where $n$ is the dimensionality of the feature space. The most commonly used parameterizations include handcrafted linear functions, state-based linear functions, and nonlinear models such as neural networks.

In handcrafted linear functions, the reward is modeled as $R(s) = \phi(s)^\top w + b$, with $w \in \mathbb{R}^n, b \in \mathbb{R}$ learnable parameters, and the feature representation $\phi(s)$ is manually designed to capture properties related to the system's objective, in our case, the resilience outcome. Relevant features may include metrics such as resource availability, fairness indicators, or inter-agent distances. This approach offers high interpretability and simplicity, but heavily depends on the quality of the handcrafted features.

In state-based linear functions, the feature representation is set as $\phi(s) = s$ directly using the raw state variables. This approach removes the need for feature engineering but may struggle to capture complex relationships when resilience-relevant properties are nonlinearly entangled in the state space. Nonlinear models, such as neural networks parameterized as $R(s, \theta)$ offer greater flexibility by capturing complex, nonlinear dependencies between the state and resilience outcomes. However, they typically involve high-dimensional parameter spaces and require larger datasets and longer training processes to achieve good generalization. Moreover, the optimization landscape becomes more challenging, increasing the risk of convergence to suboptimal local minima instead of the desired global optimum.

Once a parameterization is chosen, we formulate the preference-based learning problem using the trajectory rankings induced by cooperative resilience scores. We explore two approaches to learn a reward function from these preferences: (i) Margin-based Preference Learning (MPL), and (ii) Probabilistic Preference Learning (PPL).

### 3.2.1 Margin-based Preference Learning

This approach aims to learn a reward function such that more resilient trajectories accumulate higher total reward than less resilient ones. Given a pair of ranked trajectories $(\tau_i, \tau_j)$, where $\tau_i \succ \tau_j$ indicates that $\tau_i$ is more resilient than $\tau_j$, as measured by an external cooperative resilience metric. Then the learning objective is to ensure that $\sum_{s \in \tau_i} R(s; \theta) > \sum_{s \in \tau_j} R(s; \theta)$.

To model this preference, we introduce a margin $\delta_{ij} > 0$, which can be fixed (e.g. $\delta_{ij} = 1$, as in traditional MPL) or dynamically set to reflect the resilience gap between trajectories $\delta_{ij} = |\rho(\tau_i) - \rho(\tau_j)|$.

We then formulate the optimization problem as

$$\min_\theta \quad \sum_{(\tau_i \succ \tau_j)} \max \left( 0, \delta_{ij} - \left( \sum_{s \in \tau_i} R(s; \theta) - \sum_{s \in \tau_j} R(s; \theta) \right) \right).$$

When $R(s, \theta)$ is a linear function of state features, this problem is convex. For nonlinear models (e.g., neural networks), the objective becomes non-convex, and the optimization must be approached using iterative methods with appropriate regularization. Another important factor in optimization is

how trajectory pairs are selected, as different sampling strategies influence convergence and the type of preferences captured. We evaluated three approaches: *random*, *ranked* (adjacent in the resilience order), and *mixed* (a probabilistic combination of both). The margin-based formulation thus yields six variants, combining the two margin definitions ($\delta_{ij} = 1$ or $\delta_{ij} = |\rho(\tau_i) - \rho(\tau_j)|$) with the three sampling strategies. Full implementation details are provided in Appendix A.4.

### 3.2.2 PROBABILISTIC PREFERENCE LEARNING

In PPL, we define the probability that the trajectory $\tau_i$ is preferred over $\tau_j$ as a function of their cumulative rewards. The learning problem is then formulated as a maximum likelihood estimation over the observed preferences. Equivalently, the objective can be expressed as the minimization of the negative log-likelihood:

$$\min_\theta \quad - \sum_{(\tau_i \succ \tau_j)} \log \left( \frac{\exp \left( \sum_{s \in \tau_i} R(s; \theta) \right)}{\exp \left( \sum_{s \in \tau_i} R(s; \theta) \right) + \exp \left( \sum_{s \in \tau_j} R(s; \theta) \right)} \right).$$

This formulation is convex when $R(s, \theta)$ is a linear function of state features, and allows efficient optimization using standard convex solvers. Compared to MPL, this approach provides smooth gradients, which can lead to more stable convergence and better handling of noisy or uncertain rankings. However, it may be computationally more expensive due to the use of exponential operations over each trajectory pair. Data sampling strategies are analogous to MPL case (see Appendix A.4).

## 4 EXPERIMENTAL VALIDATION

### 4.1 ENVIRONMENT DESCRIPTION

We evaluate our approach in a simplified version of a *social dilemma* inspired by the "Commons Harvest" scenario from the Melting Pot suite Agapiou et al. (2022); Perolat et al. (2017). The original environment is defined as partially observable. In this work, we deliberately adopt a *fully observable* variant. This is consistent with many fully observable Markov games used in cooperative and mixed-motive MARL. The environment consists of a discrete $8 \times 8$ grid where 2 agents interact and harvest resources from a shared tree containing 16 apples, located in the central region of the grid (see Figure 1). Apples grow probabilistically, with regrowth chances increasing as more apples are preserved—encouraging sustainable behavior. This creates interdependence: while agents benefit from consumption, overharvesting reduces future availability and harms collective outcomes. This setting naturally captures a **mixed-motive scenario**: agents must balance individual consumption with the long-term collective benefit of preserving the resource pool.

### 4.2 TRAJECTORY COLLECTION AND RESILIENCE-BASED RANKING

To initialize the reward inference process, we collect 500 trajectories generated by agents following a random policy, each lasting 1000 steps. At timestep 500, a disruption removes apples from the central tree with fixed probability, ensuring that at least one remains so the episode can continue. These trajectories are then ranked according to their **cooperative resilience** score, computed from the indicators introduced in Subsection 3.1. The resulting ranking serves as input to the preference-based IRL algorithms described in Subsection 3.2 to infer a reward function that promotes cooperative resilience. Full details of the experimental setup and configurations are provided in the Appendix (Appendix A).

Note that the cooperative-resilience score is used to construct the trajectory rankings and, as discussed later, as one of our evaluation metrics. This does not introduce direct circularity, since agents never receive the resilience score during training and are instead guided only by the inferred reward function, which is a function of the state rather than of the trajectory-level metric.

## 4.3 REWARD INFERENCE CONFIGURATIONS

Building on the trajectory ranking, we evaluate multiple reward inference configurations to determine which best support collective well-being. We consider two main approaches: (i) resilience-based, where a unique reward function $\hat{R}$ is inferred and shared by all agents, and (ii) hybrid, where $\hat{R}$ is combined with a consumption-based individual reward. In the hybrid setting, each agent receives a reward composed of the shared resilience term and its own consumption signal.

Both approaches are tested with three parameterizations of $\hat{R}$: handcrafted, state-based linear, and neural network, and two optimization methods: Margin-based Preference Learning (MPL) and Probabilistic Preference Learning (PPL). MPL yields six variants, from two margin schemes and three sampling strategies, while PPL yields three, resulting in 27 total configurations (see Appendix A.6).

The inferred rewards are injected into PPO agents, which are trained for 500 episodes of 1000 steps and evaluated under the same disruption protocol used in trajectory ranking, where a subset of apples is removed from the central tree at step 500. Figure 2 reports the metric associated with the consumption of the last remaining resource, comparing the best configurations of each reward parameterization and both optimization approaches.

In selecting these *best* configurations, we did not rely on resilience alone. Several PPL variants achieve high resilience scores (e.g., PPL-R and PPL-K), but do so by inducing overly conservative policies with low average rewards, preserving resources at the cost of individual task performance. To avoid such undesirable behaviors, we adopted a **multi-criteria selection rule**: high resilience, high cumulative reward, low last-apple consumption, and low variance across episodes. Under this joint evaluation, the MPL–M1 Hybrid with handcrafted features consistently dominated the alternatives. We adopt this reward configuration, hereafter referred to as our ***hybrid strategy***, as the reference for comparison against baseline methods under the new disruption protocol introduced in Subsection 4.4. Appendix A.5.3 provides additional metrics and plots for all configurations. 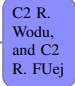

It is important to note that our chosen *hybrid strategy* relies on a handcrafted reward parameterization. Its strong performance is partly explained by the fact that its features encode meaningful prior structure about the domain, making it the least general and the most dependent on expert knowledge. At the same time, our preference-based IRL procedure must still learn appropriate weights for these features to obtain resilient, low-selfishness behavior; without the resilience-based rankings, the same features do not automatically yield suitable policies. By contrast, the linear and neural models operate directly over the full joint state and are therefore more data-hungry; with only 500 ranked trajectories in a high-dimensional, spatially structured environment, they are likely underpowered rather than fundamentally flawed.

## 4.4 EVALUATION METRICS AND EXPERIMENTAL RESULTS

We have implemented an expanded disruption protocol for evaluation, applied to agent trained with *hybrid strategy*. Evaluating algorithms under the same disruption protocol used during training may risk overfitting to known conditions, thus limiting the generalization claims of our method. To address this, the new protocol introduces three temporally distributed and qualitatively distinct disruptions, each lasting 5000 steps: (i) resource removal at step 1250, (ii) a temporary reduction in

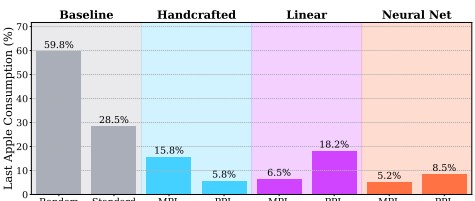
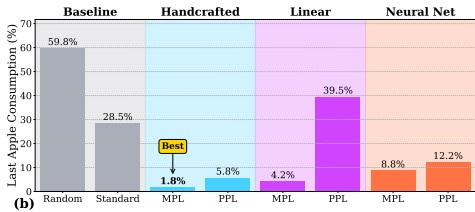

Figure 2: Percentage of episodes (out of 500) in which agents consumed the last remaining apple for the best configurations under each reward parameterization and optimization method. **(a)** Agents trained exclusively with resilience-aligned rewards. **(b)** Agents trained with *hybrid strategy*.

apple regrowth rate starting at step 2500, and (iii) an agent failure simulation, where one agent loses control and moves randomly from steps 3750 to 3900.

In these evaluations, we consider three baselines: a random policy, PPO with a standard reward scheme (+1 for consuming an apple, 0 otherwise), and QMIX. For QMIX, the individual reward function is also defined using the +1/0 scheme, but in practice this leads agents to converge toward regions without apples. To mitigate this, we increased the reward to +10 for consuming an apple. Using this modified reward, the trained QMIX agents were evaluated with $\epsilon = 0$ under the disruption protocol. These baselines are contrasted against our *hybrid strategy*, implemented as PPO with the resilience-informed reward learned through our method.

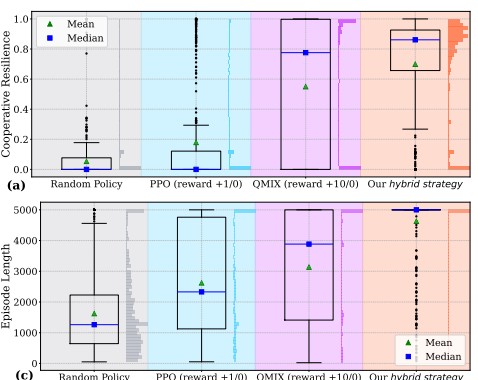
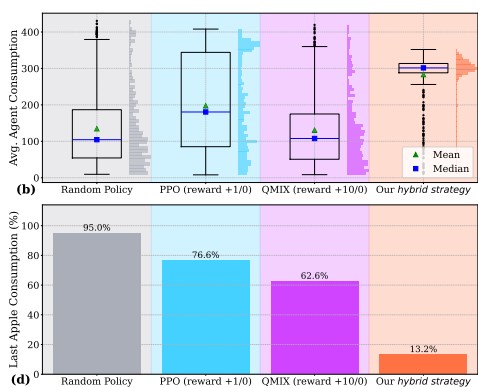

Figure 3: Performance metrics over 500 episodes. **(a)** Cooperative resilience. **(b)** Average total apple consumption per episode across both agents. **(c)** Episode length. **(d)** Last-apple consumption frequency, indicating the occurrence of social dilemma failures.

Figure 3 summarizes performance metrics across 500 evaluation episodes. Panel (a) shows that cooperative resilience is consistently higher for the *hybrid strategy*, with both mean and median values shifted toward the upper end of the scale relative to all baselines. Panel (b) indicates that *hybrid strategy* achieves the highest average cumulative consumption across agents, demonstrating that sustainability is attained without sacrificing productivity. Panel (c) further reveals that episode lengths are significantly extended: under *hybrid strategy*, resources typically remain available until the simulation horizon (5000 steps), indicating more efficient and balanced exploitation. Finally, panel (d) shows that the social dilemma of last-resource depletion is substantially mitigated: the last apple is consumed in only 13.2% of episodes.

To statistically validate these findings, we applied the Mann–Whitney U test for each metric, with $p$-values corrected using the Benjamini–Hochberg procedure (FDR, $\alpha = 0.05$). Bonferroni-adjusted $p$-values are reported in Appendix A.8.1. Results confirm that, for cooperative resilience, *hybrid strategy* significantly outperforms Random and PPO, while no significant difference is found relative to QMIX. In contrast, for both cumulative consumption and episode length, *hybrid strategy* outperforms **all** baselines after correction.

To further interpret agent behavior, we visualize the position frequency maps over 500 evaluation episodes (Figure 4). Agent 1 is shown in shades of green and Agent 2 in purple, with apple positions marked in red. Under the random policy, agents spread almost uniformly across the grid, with no clear coordination. With PPO rewards, both agents cluster in the bottom-left corner, strongly overlapping and competing for the same apples. QMIX produces an alternative but still suboptimal pattern: agents remain concentrated in the opposite corner without diversifying their movement across the grid. By contrast, the *hybrid strategy* shows a complementary specialization: Agent 1 explores a wider area, while Agent 2 remains anchored along the right boundary, harvesting resources with little movement. This emergent division of roles could avoids redundant visits and illustrates how cooperative behavior can arise from differentiated strategies. For additional visualization, Appendix C includes disaggregated position maps. These plots represent, with circles, the locations where each agent spent the most time, together with individual maps per agent to highlight their distinct spatial behaviors.

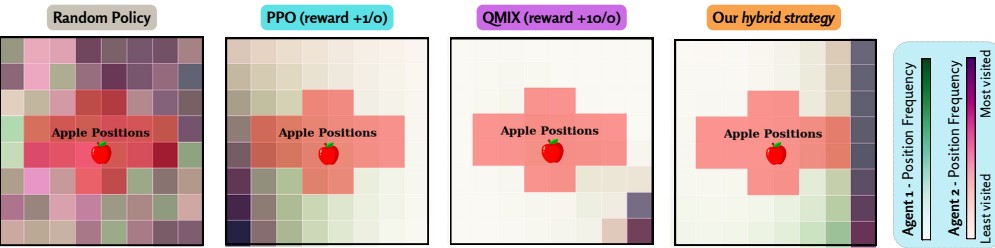

Figure 4: Position frequency maps for Agent 1 (green) and Agent 2 (purple) under four training configurations: (i) random policy, (ii) PPO with standard rewards, (iii) QMIX, and (iv) *hybrid strategy*. Each heatmap depicts the spatial visitation density over 500 evaluation episodes, with apple locations marked in red. Agents were randomly initialized at the start of each episode and evaluated under the same protocol with three disruption events.

## 4.5 EVALUATING SCALABILITY

The initial evaluation of our approach was conducted in a simplified setting with few agents. Several extensions remain to be explored, including more complex and partially observable domains. To assess the scalability of our pipeline, we implemented a larger $16 \times 16$ grid-world with 4 agents and 3 apple trees (see Appendix A.8.2). In this environment, each tree disappears permanently once all surrounding apples are harvested, introducing a localized resource depletion mechanism and stronger interdependencies among agents. Moreover, resources can only regenerate up to a much lower threshold (16 apples in total, instead of the initial distribution), effectively limiting regrowth to the equivalent of a single tree.

We applied our full pipeline in this extended environment, including the computation of resilience indicators and reward inference using *hybrid strategy*. For reward inference, we relied on 400 ranked trajectories generated from random agent behavior (see Appendix A.7). The evaluation protocol consisted of 50 episodes, each lasting 2000 steps, with a disruption introduced in 300 timestep by removing apples from the environment. In this setting, we directly transferred the *hybrid strategy* process identified in the smaller environment. Thus, the results here should be interpreted as a practical workflow in which configuration searches are performed in small environments and then the scalability of the larger ones is evaluated.

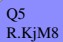

The algorithm was compared against a random policy and PPO with the traditional reward scheme. As summarized in Table 1, our *hybrid strategy* achieves higher average cooperative resilience, longer episode durations, and a reduction in social dilemma failures. To further evaluate robustness, we applied the Mann–Whitney U test with Benjamini–Hochberg corrections when comparing *hybrid strategy* against standard PPO. The results suggest that, although the improvement in average resilience is not statistically significant in this setting, resilience-based rewards during training lead to significantly longer episodes and higher cumulative rewards. These findings indicate enhanced system survivability and sustained agent performance, consistent with the intended goals of cooperative resilience. Additional results and statistical analyses for this environment are provided in Appendix A.8.2

Table 1: Comparison of algorithms in the extended $16 \times 16$ environment with four agents.

| Method | Cooperative Resilience | Apple Consumption | Episode Length | Last Apple |
|---|---|---|---|---|
| *Hybrid strategy* | $0.889 \pm 0.395$ | $22.93 \pm 4.77$ | $1923 \pm 263$ | 6 / 50 |
| PPO (reward +1/0) | $0.814 \pm 0.469$ | $16.74 \pm 4.50$ | $1450 \pm 625$ | 25 / 50 |
| Random policy | $0.274 \pm 0.293$ | $14.51 \pm 3.47$ | $760 \pm 361$ | 50 / 50 |

## 4.6 DISCUSSION OF RESULTS

Our experimental findings show that cooperative resilience can be improved through the proposed framework. Agents trained with the *hybrid strategy* outperform baseline policies, achieving higher

cooperative resilience and more structured behaviors even when trained from random demonstrations. Incorporating individual incentives through the *hybrid strategy* preserves cooperative performance, demonstrating that individual and collective goals can be successfully aligned. Notably, the handcrafted margin-based approach offers a balance between interpretability and impact.

Under this strategy and in the evaluation protocol with three interruptions, the percentage of episodes with last-apple consumption, a proxy for selfishness, drops to only 13.2%, while maintaining apple consumption levels higher than those of the baseline methods. This confirms that agents can achieve task effectiveness while promoting fair and prosocial behavior. Moreover, spatial maps further reveal that the *hybrid strategy* promotes complementary behaviors: one agent explores the environment while the other remains anchored along the boundary, consistently harvesting resources and supporting sustainability. These patterns are consistent with the handcrafted reward design, which incentivizes proximity avoidance and presence in resource-rich areas without direct competition (see Appendix B). An illustrative example is provided in the supplementary video.[1]

Our comparisons are restricted to random policy, PPO and QMIX, leaving out more recent cooperative MARL baselines. Nonetheless, we emphasize that the proposed contribution is not a new MARL algorithm *per se*, but rather a reward learning framework for identifying useful incentives in mixed-motive settings. Such rewards could in principle be integrated into a wide range of existing MARL methods. Regarding scalability, we acknowledge that fully scaling to larger populations remains a central direction for future work. Scaling the framework to more complex environments will likely incur higher computational costs, suggesting caution when generalizing our findings and motivating further exploration under more realistic and scalable conditions. 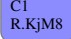 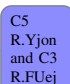

## 5 CONCLUSION AND FURTHER WORK

We introduced a framework for learning reward functions from ranked trajectories using a cooperative resilience metric in mixed-motive environments. By inferring a collective reward component and integrating it into MARL training, our method consistently improved system-level performance over baselines such as PPO and QMIX with traditional rewards. Agents trained with our pipeline sustained functionality under disruption, achieved longer episodes, higher cumulative consumption, and stronger cooperative resilience. Moreover, these agents developed structured spatial behaviors that reflect an alignment between incentives and collective outcomes. Scalability tests in larger environment further confirmed these benefits, underscoring the potential of our reward design as a principled approach that can complement existing MARL methods.

This work opens several directions for future research. First, extending the framework to the full version of the social dilemma environment under partial observability, as commonly encountered by agents, would provide a more realistic testbed for cooperative resilience. Future work should also explore larger and continuous state spaces, as well as adversarial settings, to evaluate the robustness of the approach under more demanding conditions. We acknowledge that fully scaling to more agents and to additional Melting Pot scenarios remains an important direction for future work. Another promising direction is to expand the set of baselines, incorporating recent cooperative MARL algorithms. Since our contribution is not a new algorithm but a method for inferring useful rewards, these inferred incentives could be integrated into diverse MARL approaches to analyze how resilience-oriented signals shape their performance. 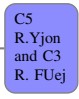

Additionally, integrating human-generated trajectories as preference signals may offer valuable insights into natural resilience strategies, enabling the design of artificial agents that better align with human values and cooperative norms. This human-in-the-loop perspective could strengthen the applicability of cooperative resilience learning in real-world settings.

## REPRODUCIBILITY STATEMENT

Source code and experimental configurations will be released in an anonymized GitHub repository to ensure reproducibility (see Appendix F). The main paper describes the environments, reward inference methods, and evaluation protocols. Additional implementation details, extended results,

---

[1]See supplementary video: `https://youtu.be/4AdLDhyKqKY`

and statistical analyses are provided in the Appendix. A copy of the source code is also included as supplementary material as a compressed `.zip` file.

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

## A EXPERIMENTAL SETUP DETAILS

In this section we provide additional details for reproducing the experiments described in Section 4. This includes hyperparameters, environment dynamics, and disruption protocols. All experiments were conducted using Google Colab with access to standard GPUs (e.g., NVIDIA T4) and 12–16 GB of RAM.

- **Episodes:** Each evaluation was performed over 500 episodes.
- **Disruption:** Each episode consisted of 1000 timesteps, with a disruption consistently introduced at timestep 500.
- **Agents:** All experiments use 2 agents. Scalability to more agents is discussed as limitations due to computational costs.
- **Learning:** PPO was used as the underlying RL algorithm.
- **Framework variants:** Three types of rewards were tested — standard individual reward, resilience-inferred reward, and a hybrid of both.

### A.1 PPO AGENT ARCHITECTURE

The PPO agent used in our experiments is implemented with a shared-encoder Actor-Critic architecture. The model, defined by the `ActorCritic` class, consists of:

- A fully connected input layer with 128 hidden units and ReLU activation.
- An actor head that outputs a probability distribution over discrete actions via a softmax activation.
- A critic head that estimates the scalar state value.

During inference, the model outputs both the action probabilities and a value estimate for a given state.

**Agent Configuration:** The `PPOAgent` class wraps this architecture and includes the training logic:

- **Optimizer:** Adam, learning rate of $1 \times 10^{-5}$.
- **Discount factor:** $\gamma = 0.99$.
- **Clipping threshold:** $\epsilon = 0.01$ for the PPO surrogate objective.
- **Replay buffer:** A deque storing up to 1000 transitions, each as a tuple $(\text{state}, \text{action}, \text{reward}, \text{next state}, \text{old probability})$.

### A.2 PPO TRAINING PROCEDURE

Training is triggered once the buffer reaches at least 200 transitions. The loss function combines: (i) The clipped PPO surrogate objective using the ratio between new and old action probabilities. (ii) A value loss term, implemented as the squared advantage.

The final loss is computed as:

$$\mathcal{L} = -\min(r_t A_t, \text{clip}(r_t, 1 - \epsilon, 1 + \epsilon) A_t) + 0.5 \cdot A_t^2$$

where:

- $r_t = \frac{\pi(a_t|s_t)}{\pi_{\text{old}}(a_t|s_t)}$ is the probability ratio between the new and old policy for the taken action.
- $A_t = R_t - V(s_t)$ is the advantage, computed as the difference between the observed return $R_t$ and the critic's value estimate $V(s_t)$.
- The first term ensures conservative policy updates using the clipped surrogate objective.
- The second term is the squared advantage, acting as a value loss.

This implementation does not use Generalized Advantage Estimation (GAE) or entropy regularization, favoring simplicity and efficiency for compact environments.

## A.3 DISRUPTION PROTOCOLS

- **Trajectories for reward learning.**

  To generate the trajectories used in the reward inference process, we introduced a single disruption event at timestep $t = 500$ of each episode (episodes lasted 1000 timesteps). This disruption simulated a sudden and unexpected loss of resources in the environment.

  Specifically, a fixed proportion $(40\%)$ of the apples present in the grid at the moment of disruption was randomly removed. To avoid trivial cases, this removal was only applied when more than one apple was available.

- **Test environment for evaluation.**

  To evaluate the generalization and adaptive capacity of the agents under unseen adverse conditions and to avoid overfitting to the single disruption trajectories used during training, we designed a more challenging disruption protocol with three distinct disruptions per episode. Each test episode lasted 5000 timesteps, and three disruptions were distributed temporally as follows:

  - **(1) Resource removal.** At step $t = 1250$, we applied the same partial removal mechanism as in training (random deletion of $40\%$ of apples currently present).
  - **(2) Regrowth rate reduction.** From steps $t = 2500$ to $t = 2600$, the apple regrowth dynamics were altered. In the standard environment, At each timestep $t$, the probability of regrowth for an available site is given by

$$p_{\text{regen}}(t) = r \frac{A(t)}{\tau}, \tag{1}$$

  where $r$ is the base regeneration rate fixed at $0.05$ in the original environment, $A(t)$ is the number of apples currently present in the grid, and $\tau$ is the saturation threshold. During the disruption, the rate was reduced to $r = 0.01$, causing a temporary slowdown in resource replenishment.

  - **(3) Agent failure.** From steps $t = 3750$ to $t = 3900$, both agents temporarily lost control and executed purely random actions, simulating a failure or malfunction scenario.

Figure A.1 illustrates the temporal distribution of disruption events in both training and testing settings. It highlights the occurrence of each disruption and the corresponding failure and recovery windows.

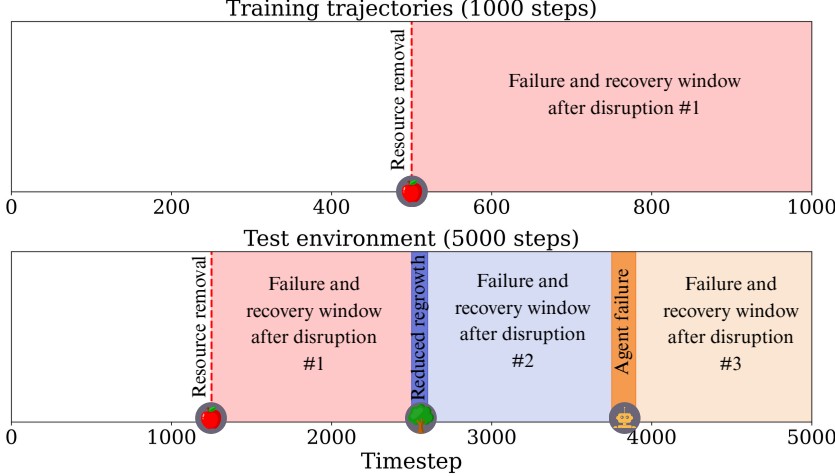

Figure A.1: Timeline of disruption events used in training (top) and testing (bottom).

## A.4 DATA SAMPLING STRATEGIES

In preference-based reward learning, the optimization problem is solved iteratively by presenting trajectory pairs to the optimizer. The way these pairs are sampled strongly affects convergence and the type of preferences captured. We consider three strategies:

- **Random Sampling**: trajectory pairs are drawn uniformly at random. This is akin to stochastic gradient descent and introduces noise that may help escape local minima.
- **Ranked Sampling**: trajectory pairs are selected from adjacent trajectories in the resilience ranking, emphasizing fine-grained preference boundaries.
- **Mixed Sampling**: with probability $p$, a close-ranked pair is selected; otherwise, with probability $1-p$, a random pair is used. This balances exploration with sensitivity to the ranking structure. In our experiments, we set $p = 0.9$.

## A.5 EVALUATION LEARNING CONFIGURATIONS

To analyze the performance of different reward learning strategies, we evaluate a combination of reward objectives, reward parameterizations, and preference learning losses.

- **Reward Objectives:** We compare two types of learned reward functions:
  1. Resilience-based reward, this reward is inferred exclusively from ranked agent trajectories using a cooperative resilience metric, without relying on any direct task or consumption signal.
  2. Hybrid, which balances individual consumption incentives with resilience-based reward.
- **Reward Parameterizations:** Each reward function is trained using one of the following models:
  1. Handcrafted (linear weights over manually designed features),
  2. Linear (using raw state vector as input),
  3. Neural Network (nonlinear function over the raw state).
- **Preference Learning Methods:** We compare Margin-based Preference Learning and Probabilistic Preference Learning models to infer the reward from ranked trajectories.

Each configuration is evaluated over 6 simulations for Margin-based Preference Learning (with variations in sampling and margin values) and 3 simulations for Probabilistic Preference Learning (with different sampled rankings), yielding a total of 54 experimental runs.

In the following, we present quantitative results for each configuration, including: Mean and standard deviation of the resilience metric. Mean and variability of agent rewards (individual and average). Percentage of episodes with the social dilemma outcome (Last Apple consumption). These results are grouped by reward objective (Resilience-only or Hybrid) and organized per reward parameterization (handcrafted, linear, neural network).

The following nomenclature is used throughout the tables for clarity and consistency.

**Baseline Configurations:**

- **Random:** Agents select actions uniformly at random.
- **Standard:** PPO agents trained with a standard individual reward based on apple consumption, without preference learning or inferred rewards.

**Margin-based Preference Learning Configurations:**

- **MPL-R1:** Margin-based Preference Learning with random trajectory sampling and a fixed margin of 1.
- **MPL-Rk:** Margin-based Preference Learning with random sampling and a variable margin $k$, computed as the cooperative resilience difference between the two trajectories in the ranked pair.

- **MPL-K1:** Margin-based Preference Learning with k-best sampling and margin = 1.
- **MPL-Kk:** Margin-based Preference Learning with k-best sampling and margin = resilience difference ($k$).
- **MPL-M1:** Margin-based Preference Learning with mixture sampling (random + k-best) and margin = 1.
- **MPL-Mk:** Margin-based Preference Learning with mixture sampling and margin = resilience difference.

**Probabilistic Preference Learning Configurations:**

- **PPL-R:** Probabilistic Preference Learning model with random sampling.
- **PPL-K:** Probabilistic Preference Learning with k-best sampling.
- **PPL-M:** Probabilistic Preference Learning with mixture sampling.

**Column headings used across tables:**

- **Config:** Name of the training configuration or baseline.
- **Res.:** Mean resilience score.
- **AvgR:** Average reward across both agents.
- **R1 / R2:** Individual rewards for Agent 1 and Agent 2.
- **Res. SD / AvgR SD / R1 SD / R2 SD:** Standard deviations over episodes, reflecting variability.
- **Last %:** Percentage of episodes in which the last apple was consumed (indicator of social dilemma severity).

**Highlighting:** Values highlighted in red indicate best performance within a given table (e.g., highest resilience, lowest variability, or lowest last-apple occurrence) for each loss function.

### A.5.1 RESILIENCE-BASED REWARD RESULTS

This section reports the results of resilience-based rewards, where the inferred reward is derived solely from cooperative resilience. The same reward function is shared identically across all agents.

Table A.2: Performance metrics under handcrafted reward parametrization.

| Config | Res. | AvgR | R1 | R2 | Res. SD | AvgR SD | R1 SD | R2 SD | Last % |
|--------|------|------|------|------|---------|---------|-------|-------|--------|
| Random | 0.74 | 62.98 | 63.80 | 62.16 | 0.49 | 25.12 | 27.15 | 26.66 | 59.75 |
| Standard | 0.85 | 66.37 | 83.32 | 49.41 | 0.39 | 19.29 | 26.95 | 16.80 | 28.50 |
| MPL-R1 | 0.88 | 69.21 | 67.33 | 71.09 | 0.36 | 18.05 | 21.11 | 21.10 | 24.25 |
| MPL-Rk | 0.85 | 67.73 | 72.50 | 62.96 | 0.39 | 20.25 | 24.24 | 21.67 | 29.50 |
| MPL-K1 | 0.79 | 64.87 | 72.33 | 57.42 | 0.49 | 24.19 | 28.52 | 23.24 | 41.75 |
| MPL-Kk | 0.86 | 67.14 | 69.47 | 64.81 | 0.41 | 20.47 | 24.41 | 22.17 | 30.75 |
| MPL-M1 | 0.77 | 65.46 | 67.92 | 63.00 | 0.49 | 23.50 | 26.08 | 25.04 | 46.00 |
| MPL-Mk | 0.95 | 70.97 | 66.10 | 75.85 | 0.26 | 14.47 | 16.94 | 19.20 | 15.75 |
| PPL-R | 0.97 | 58.37 | 58.82 | 57.92 | 0.17 | 10.11 | 14.18 | 14.77 | 5.75 |
| PPL-K | 0.97 | 58.15 | 72.25 | 44.05 | 0.15 | 9.23 | 15.23 | 12.98 | 5.75 |
| PPL-M | 0.84 | 67.36 | 61.18 | 73.54 | 0.42 | 21.12 | 21.34 | 25.55 | 36.25 |

Table A.3: Performance metrics under linear reward parametrization.

| Config | Res. | AvgR | R1 | R2 | Res. SD | AvgR SD | R1 SD | R2 SD | Last % |
|---|---|---|---|---|---|---|---|---|---|
| Random | 0.74 | 62.98 | 63.80 | 62.16 | 0.49 | 25.12 | 27.15 | 26.66 | 59.75 |
| Standard | 0.85 | 66.37 | 83.32 | 49.41 | 0.39 | 19.29 | 26.95 | 16.80 | 28.50 |
| MPL-R1 | 0.86 | 66.94 | 70.67 | 63.22 | 0.37 | 19.47 | 23.09 | 20.81 | 31.25 |
| MPL-Rk | 0.97 | 63.53 | 105.24 | 21.82 | 0.16 | 10.41 | 18.51 | 9.23 | 6.50 |
| MPL-K1 | 0.94 | 61.51 | 60.98 | 62.03 | 0.25 | 13.30 | 16.88 | 17.63 | 11.50 |
| MPL-Kk | 0.90 | 66.55 | 61.17 | 71.94 | 0.33 | 17.43 | 19.85 | 22.18 | 19.50 |
| MPL-M1 | 0.94 | 71.41 | 91.79 | 51.03 | 0.25 | 14.41 | 21.86 | 14.54 | 14.00 |
| MPL-Mk | 0.96 | 62.28 | 93.21 | 31.34 | 0.18 | 11.49 | 20.36 | 10.47 | 7.75 |
| PPL-R | 0.86 | 67.16 | 50.46 | 83.86 | 0.39 | 19.04 | 16.93 | 26.94 | 26.25 |
| PPL-K | 0.89 | 68.23 | 81.09 | 55.37 | 0.37 | 19.42 | 25.67 | 19.05 | 24.25 |
| PPL-M | 0.92 | 70.13 | 42.04 | 98.22 | 0.32 | 16.69 | 12.97 | 24.99 | 18.25 |

Table A.4: Performance metrics under neural network reward parametrization.

| Config | Res. | AvgR | R1 | R2 | Res. SD | AvgR SD | R1 SD | R2 SD | Last % |
|---|---|---|---|---|---|---|---|---|---|
| Random | 0.74 | 62.98 | 63.80 | 62.16 | 0.49 | 25.12 | 27.15 | 26.66 | 59.75 |
| Standard | 0.85 | 66.37 | 83.32 | 49.41 | 0.39 | 19.29 | 26.95 | 16.80 | 28.50 |
| MPL-R1 | 0.89 | 68.20 | 83.56 | 52.85 | 0.35 | 17.71 | 24.03 | 16.63 | 26.25 |
| MPL-Rk | 0.97 | 55.27 | 64.17 | 46.37 | 0.18 | 10.28 | 16.23 | 12.24 | 5.25 |
| MPL-K1 | 0.92 | 69.36 | 79.54 | 59.18 | 0.31 | 15.81 | 21.96 | 17.27 | 14.50 |
| MPL-Kk | 0.83 | 65.00 | 56.58 | 73.41 | 0.41 | 21.77 | 22.59 | 26.45 | 36.50 |
| MPL-M1 | 0.95 | 62.74 | 85.75 | 39.73 | 0.24 | 12.24 | 19.56 | 12.62 | 9.75 |
| MPL-Mk | 0.82 | 66.93 | 80.42 | 53.43 | 0.44 | 22.16 | 29.68 | 20.19 | 34.00 |
| PPL-R | 0.93 | 68.87 | 59.01 | 78.74 | 0.32 | 16.04 | 17.54 | 21.35 | 16.00 |
| PPL-K | 0.96 | 65.83 | 71.43 | 60.22 | 0.20 | 12.09 | 17.42 | 15.76 | 8.50 |
| PPL-M | 0.91 | 70.46 | 55.20 | 85.72 | 0.34 | 18.52 | 17.87 | 26.74 | 23.00 |

## A.5.2 HYBRID REWARD RESULTS

This section presents the results of hybrid rewards, where each agent receives the traditional consumption-based reward (+1 for consuming an apple, 0 otherwise) combined with the previously inferred resilience-based reward.

Table A.5: Performance metrics under handcrafted reward parametrization hybrid reward.

| Config | Res. | AvgR | R1 | R2 | Res. SD | AvgR SD | R1 SD | R2 SD | Last % |
|---|---|---|---|---|---|---|---|---|---|
| Random | 0.74 | 62.98 | 63.80 | 62.16 | 0.49 | 25.12 | 27.15 | 26.66 | 59.75 |
| Standard | 0.85 | 66.37 | 83.32 | 49.41 | 0.39 | 19.29 | 26.95 | 16.80 | 28.50 |
| MPL-R1 | 0.91 | 68.93 | 48.41 | 89.45 | 0.32 | 16.25 | 15.59 | 23.83 | 18.00 |
| MPL-Rk | 0.79 | 66.41 | 65.97 | 66.84 | 0.45 | 23.40 | 26.69 | 24.77 | 42.75 |
| MPL-K1 | 0.88 | 68.10 | 79.09 | 57.11 | 0.41 | 20.32 | 26.84 | 19.69 | 27.25 |
| MPL-Kk | 0.82 | 64.29 | 69.49 | 59.09 | 0.43 | 21.84 | 25.28 | 23.31 | 33.25 |
| MPL-M1 | 0.99 | 62.88 | 89.80 | 35.96 | 0.09 | 7.68 | 15.23 | 9.71 | 1.75 |
| MPL-Mk | 0.91 | 70.80 | 58.30 | 83.29 | 0.33 | 18.10 | 17.89 | 24.30 | 27.00 |
| PPL-R | 0.86 | 67.98 | 68.59 | 67.37 | 0.40 | 20.72 | 23.34 | 23.12 | 30.75 |
| PPL-K | 0.96 | 59.43 | 88.85 | 30.00 | 0.18 | 10.61 | 17.11 | 10.50 | 5.75 |
| PPL-M | 0.94 | 67.49 | 46.13 | 88.85 | 0.26 | 13.61 | 13.41 | 20.70 | 12.25 |

Table A.6: Performance metrics under linear reward parametrization hybrid reward.

| Config | Res. | AvgR | R1 | R2 | Res. SD | AvgR SD | R1 SD | R2 SD | Last % |
|--------|------|------|------|------|---------|---------|-------|-------|--------|
| Random | 0.74 | 62.98 | 63.80 | 62.16 | 0.49 | 25.12 | 27.15 | 26.66 | 59.75 |
| Standard | 0.85 | 66.37 | 83.32 | 49.41 | 0.39 | 19.29 | 26.95 | 16.80 | 28.50 |
| MPL-R1 | 0.87 | 68.23 | 67.71 | 68.75 | 0.41 | 21.66 | 23.54 | 24.69 | 33.25 |
| MPL-Rk | 0.83 | 66.98 | 72.26 | 61.69 | 0.42 | 21.25 | 25.70 | 22.31 | 32.25 |
| MPL-K1 | 0.99 | 61.68 | 83.97 | 39.39 | 0.14 | 9.70 | 16.23 | 13.05 | 4.25 |
| MPL-Kk | 0.94 | 62.52 | 52.83 | 72.21 | 0.31 | 14.76 | 16.37 | 18.78 | 16.75 |
| MPL-M1 | 0.94 | 67.84 | 65.48 | 70.20 | 0.35 | 16.72 | 19.11 | 21.31 | 23.25 |
| MPL-Mk | 0.89 | 69.13 | 76.16 | 62.10 | 0.35 | 18.36 | 22.38 | 19.54 | 25.75 |
| PPL-R | 0.81 | 65.90 | 68.47 | 63.33 | 0.44 | 22.81 | 25.81 | 24.04 | 36.75 |
| PPL-K | 0.83 | 66.17 | 72.93 | 59.41 | 0.45 | 23.53 | 29.04 | 22.61 | 39.50 |
| PPL-M | 0.82 | 64.24 | 65.43 | 63.05 | 0.43 | 21.23 | 23.92 | 23.60 | 35.50 |

Table A.7: Performance metrics under neural network reward parametrization hybrid reward.

| Config | Res. | AvgR | R1 | R2 | Res. SD | AvgR SD | R1 SD | R2 SD | Last % |
|--------|------|------|------|------|---------|---------|-------|-------|--------|
| Random | 0.74 | 62.98 | 63.80 | 62.16 | 0.49 | 25.12 | 27.15 | 26.66 | 59.75 |
| Standard | 0.85 | 66.37 | 83.32 | 49.41 | 0.39 | 19.29 | 26.95 | 16.80 | 28.50 |
| MPL-R1 | 0.84 | 67.70 | 69.66 | 65.74 | 0.37 | 20.12 | 23.19 | 21.82 | 30.25 |
| MPL-Rk | 0.83 | 63.99 | 71.88 | 56.11 | 0.41 | 21.68 | 25.97 | 21.08 | 40.75 |
| MPL-K1 | 0.96 | 57.55 | 70.78 | 44.32 | 0.23 | 12.38 | 16.98 | 14.44 | 8.75 |
| MPL-Kk | 0.87 | 67.82 | 76.37 | 59.27 | 0.38 | 19.71 | 24.02 | 20.31 | 26.75 |
| MPL-M1 | 0.94 | 67.80 | 37.22 | 98.39 | 0.26 | 14.88 | 12.51 | 23.93 | 13.00 |
| MPL-Mk | 0.94 | 65.63 | 85.25 | 46.02 | 0.31 | 15.23 | 22.53 | 15.35 | 19.50 |
| PPL-R | 0.95 | 67.58 | 79.66 | 55.50 | 0.24 | 13.24 | 19.32 | 14.70 | 12.25 |
| PPL-K | 0.88 | 67.35 | 58.51 | 76.19 | 0.34 | 18.13 | 19.04 | 22.88 | 27.25 |
| PPL-M | 0.88 | 68.47 | 84.19 | 52.75 | 0.35 | 17.75 | 25.36 | 17.47 | 25.50 |

### A.5.3 BEST-PERFORMING CONFIGURATIONS

In this subsection, we report the cooperative resilience metrics and average total apple consumption per episode (across agents) for the best-performing configurations of both previous strategies. These results were obtained in the training environment, which served as the basis for selecting the final models. By including them here, we provide a complete picture of the comparative performance that guided our choice of the most effective learning configurations.

Figure A.2 summarizes cooperative resilience under both reward strategies. In (a) we found that resilience-based rewards achieve high cooperative resilience scores, with distributions that are more concentrated and stable than those of the baselines, despite being inferred from random demonstrations. In (b) we show that hybrid strategies also maintains strong resilience across configurations, with handcrafted and neural parameterizations under margin-based learning yielding. Overall, these findings indicate that both strategies can produce resilience-aligned behaviors comparable to those obtained with standard PPO training.

Promoting cooperative resilience must not come at the cost of individual effectiveness. To verify that the learned reward functions preserve task completion, we measure the average total apple consumption per episode, aggregated across both agents. Figure A.3 shows the consumption distributions for both reward strategies. Panel (a) corresponds to resilience-aligned rewards, while Panel (b) shows the hybrid strategy. Across parameterizations and learning methods, average total apple consumption per episode remains broadly comparable to the baseline PPO agents.

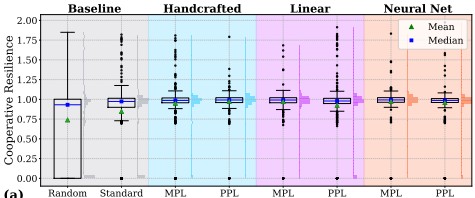 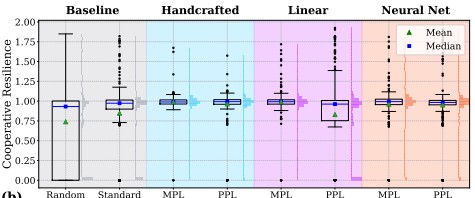

Figure A.2: Cooperative resilience over 500 simulations for the best configurations under each reward parameterization and optimization method. **(a)** Agents trained exclusively with resilience-aligned rewards. **(b)** Agents trained with hybrid rewards.

The handcrafted Margin-based Preference Learning configuration in the hybrid strategy, which exhibited the lowest incidence of selfish behavior (last-apple consumption), also maintains stable apple consumption. These results suggest that resilience-oriented incentives can support cooperative behaviors without substantially compromising individual effectiveness.

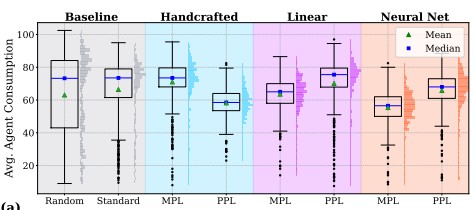 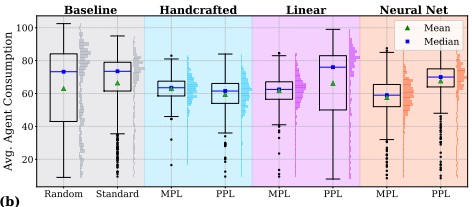

Figure A.3: Average total apple consumption per episode over 500 simulations, aggregated across both agents. **(a)** Agents trained exclusively with resilience-aligned rewards. **(b)** Agents trained with hybrid rewards.

## A.6 REWARD PARAMETRIZATION DETAILS

This subsection describes the key characteristics of the three reward function parametrizations used in our framework: handcrafted, linear, and neural network models. Each parametrization differs in its expressive capacity, interpretability, and reliance on state representations.

### A.6.1 HANDCRAFTED FEATURES

The handcrafted reward parametrization is based on six interpretable features extracted from the environment state vector $s = [x_1, x_2, y_1, y_2, m_0, m_1, ..., m_{63}]$, where $(x_1, x_2)$ and $(y_1, y_2)$ denote the positions of Agent 1 and Agent 2, and $m_i \in \{0, 1\}$ indicates the presence of an apple at cell $i$ in the $8 \times 8$ grid.

We define a feature vector $\phi(s) = [\phi_1, \phi_2, \phi_3, \phi_4, \phi_5, \phi_6]$ as follows:

- $\phi_1$ (**Remaining Apples**): Total number of apples currently present in the environment. $\phi_1 = \sum_{i=0}^{63} m_i$.

- $\phi_2$ (**Agent 1 Proximity**): Euclidean distance from Agent 1 to the nearest apple.

- $\phi_3$ (**Agent 2 Proximity**): Euclidean distance from Agent 2 to the nearest apple.

- $\phi_4$ (**Proximity Difference**): Absolute difference between the nearest-apple distances of both agents, i.e., $|\phi_2 - \phi_3|$. This feature quantifying potential asymmetries in immediate access to resources.

- $\phi_5$ (**Local Apple Density (Agent 1)**): Number of apples located in the $3 \times 3$ neighborhood around Agent 1, measuring localized resource density.

- $\phi_6$ (**Local Apple Density (Agent 2)**): Number of apples in the $3 \times 3$ neighborhood around Agent 2.

These features were selected to balance spatial information, resource availability, and local context, allowing the reward function to encode both global and local incentives relevant to cooperative behavior.

### A.6.2 LINEAR PARAMETRIZATION

In this case, the reward function is a linear model over the full flattened state vector, which includes agent positions and the status of each grid cell. The state is the vector $s = [x_1, x_2, y_1, y_2, m_0, m_1, ..., m_{63}]$, where $(x_1, x_2)$ and $(y_1, y_2)$ denote the positions of Agent 1 and Agent 2, and $m_i \in \{0, 1\}$ indicates the presence of an apple at cell $i$ in the $8 \times 8$ grid. This parametrization does not rely on feature engineering and instead learns directly from raw inputs. Although less interpretable than handcrafted models, it enables broader generalization by considering all available state information in a uniform representation.

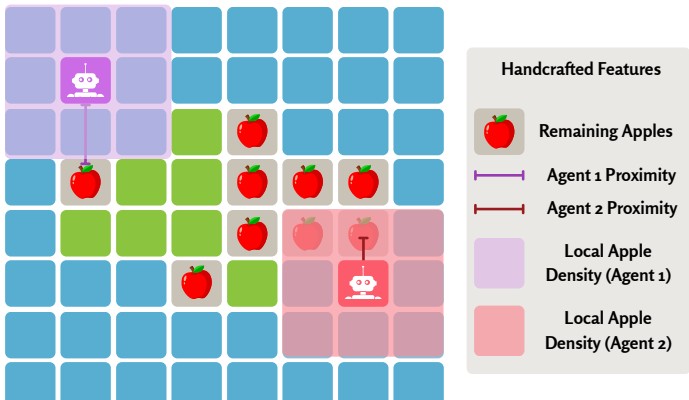

Figure A.4: Diagram of an example of handcrafted features used in the reward function. Agents (purple and red) estimate their proximity to the nearest apple, and the number of apples in their local $3 \times 3$ neighborhood.

Figure A.4 presents an illustrative example of the handcrafted features extracted from a specific grid configuration. In this case:

- $\phi_1 = 9$ corresponds to the total number of apples (in red) present on the grid.
- $\phi_2 = 1$ represents the distance from Agent 1 (top-left, purple) to its nearest apple, directly below.
- $\phi_3 = 0$ denotes the distance from Agent 2 (bottom-right, red) to its closest apple, also directly above.
- $\phi_4 = |\phi_2 - \phi_3| = |1 - 0| = 1$ captures the absolute difference in proximity to the nearest apples. In this case, Agent 2 is directly on top of an apple while Agent 1 is one cell away, indicating a mild asymmetry in immediate access to resources.
- $\phi_5 = 0$ is the number of apples within the $3 \times 3$ neighborhood centered around Agent 1.
- $\phi_6 = 2$ is the number of apples surrounding Agent 2 in its $3 \times 3$ area.

The value of $\phi_4$ is particularly informative as it reflects potential asymmetries in resource access. A low value suggests similar conditions for both agents, whereas a higher value would indicate that one agent has a significant advantage over the other in terms of proximity to resources. This feature can serve as a proxy for fairness or potential conflict over shared resources.

### A.6.3 NEURAL NETWORK PARAMETRIZATION

The neural reward function is implemented as a fully connected feedforward neural network with the following architecture:

- Input layer of size equal to the state vector dimension (`state_dim`), which includes both agent positions and the full flattened grid.

- For **Margin-based Preference Learning**, the network uses a hidden layer with **32 units** and ReLU activation.

- For **Probabilistic Preference Learning**, the network uses a hidden layer with **64 units** and ReLU activation.

- Output layer with a single unit, representing the scalar reward.

The network is optimized using the Adam optimizer with a learning rate of `0.001`. The loss function depends on the preference learning method (Margin-based Preference Learning or Probabilistic Preference Learning). This architecture was selected to provide a balance between expressive power and training efficiency, enabling the model to learn non-linear reward functions directly from raw environmental states without requiring handcrafted features.

## A.7 DETAILS ON TRAJECTORY RANKING PROCESS

To construct the ranked dataset used for reward inference, we generated a set of **400 trajectories** using agents acting under a random policy. Each trajectory consists of **1000 timesteps**, with a **disruption event triggered at timestep 500** following the protocol described in Section A.3.

Each trajectory was evaluated using the cooperative resilience metrics detail in the main paper. These metrics include:

- **Accumulated Apple Consumption (Agent 1)**: total number of apples consumed by Agent 1 over the episode.

- **Accumulated Apple Consumption (Agent 2)**: same metric for Agent 2.

- **Apples Alive**: total number of apples that remained alive in the environment at the end of the episode, reflecting sustainability.

- **Equality Index**: a fairness measure indicating how evenly the apples were distributed between both agents.

- **Collective Hunger Index**: a system-level measure aggregating the number of timesteps both agents went without eating.

These trajectories were ranked according to their **cooperative resilience score**, a scalar metric combining the previous indicators to quantify the ability to anticipate, prepare for, resist, recover from, and transform in the face of disruptive events that threaten the system joint welfare.

Figure A.5 shows in (a) the distribution of cooperative resilience scores across the 400 trajectories, using a boxplot. Additionally, Figure A.5 present in (b) the distributions of final apple consumption for each agent.

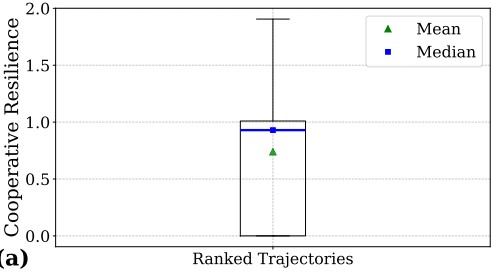 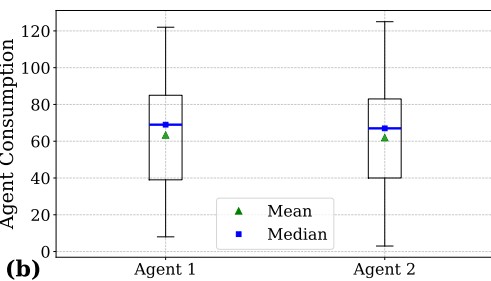

Figure A.5: Distribution of metrics in ranked trajectories. **(a)** Distribution of cooperative resilience scores. **(b)** Distribution of agent consumption.

For the 16×16 grid environment with 3 apple trees and 4 agents, we generated 400 trajectories under a random policy. Each trajectory spans 1000 timesteps, with a disruptive event consisting of apple elimination introduced at timestep 200. As in the previous environment, we used accumulated apple consumption by both agents, the number of apples remaining in the simulation, the equality index, and the collective hunger index to measure resilience. Figure A.6 in (a) shows the distribution of

cooperative resilience scores across the 400 trajectories using a boxplot. In Figure A.6 panel (b) we present the cumulative apple consumption of both agents.

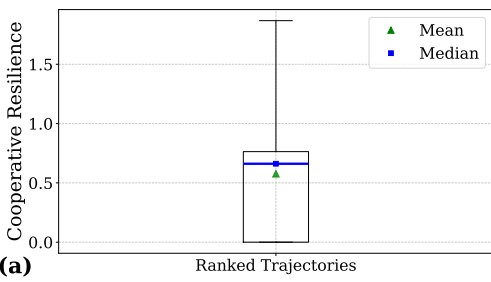 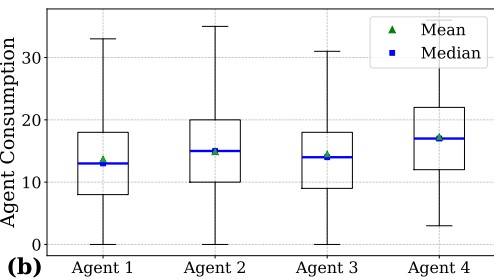

Figure A.6: Distribution of metrics in ranked trajectories in 16×16 environment. **(a)** Distribution of cooperative resilience scores. **(b)** Distribution of agent consumption.

### A.8    STATISTICAL SIGNIFICANCE TESTS

#### A.8.1    EVALUATION PROTOCOL

The system evaluation was conducted under the extended disruption protocol described in Appendix A.3. Across 500 test episodes, we assessed three metrics: cooperative resilience, average cumulative agent consumption, and episode length. For each metric, we performed pairwise statistical comparisons between *hybrid strategy* and the baseline methods: Random policy, PPO with the standard reward scheme, and QMIX.

Statistical significance was tested using the Mann–Whitney U test, a non-parametric test suitable for comparing two independent samples. To control for multiple hypotheses, we applied the Benjamini–Hochberg procedure (FDR, $\alpha = 0.05$) within each family of comparisons (per metric). For completeness, Bonferroni-corrected $p$-values are also reported. The detailed results are summarized in Tables A.8, A.9, and A.10 for cooperative resilience, average agent consumption, and episode length, respectively. In the tables, stars indicate significance levels (* $p < 0.05$, ** $p < 0.01$, *** $p < 0.001$).

The statistical analysis confirms that *hybrid strategy* significantly outperforms both the random policy baseline and PPO in terms of cooperative resilience ($p < 10^{-60}$ after corrections). No significant difference is observed against QMIX, suggesting that both methods capture cooperative patterns of similar strength. The results for average agent consumption reinforce this finding: *hybrid strategy* significantly outperforms all three baselines, including QMIX ($p < 10^{-80}$ after corrections). Finally, the results for episode length confirm that *hybrid strategy* produces significantly longer episodes than all baselines ($p < 10^{-50}$ after corrections).

Table A.8: Pairwise comparisons of cooperative resilience. Reported $p$-values from Mann–Whitney U tests, adjusted within each family of three tests using Benjamini–Hochberg (BH) and Bonferroni.

| Comparison | $p_{\text{raw}}$ | $p_{\text{BH}}$ | $p_{\text{Bonf.}}$ | Sig. |
|---|---|---|---|---|
| *Hybrid strategy* vs Random Policy | $3.53 \times 10^{-96}$ | $1.06 \times 10^{-95}$ | $1.06 \times 10^{-95}$ | *** |
| *Hybrid strategy* vs PPO (reward +1/0) | $1.38 \times 10^{-62}$ | $2.07 \times 10^{-62}$ | $4.14 \times 10^{-62}$ | *** |
| *Hybrid strategy* vs QMIX (reward +1/0) | $1.40 \times 10^{-1}$ | $1.40 \times 10^{-1}$ | $4.21 \times 10^{-1}$ | n.s. |

Table A.9: Pairwise comparisons of Agent Consumption. Reported $p$-values from Mann–Whitney U tests, adjusted within each family of three tests using Benjamini–Hochberg (BH) and Bonferroni.

| Comparison | $p_{\text{raw}}$ | $p_{\text{BH}}$ | $p_{\text{Bonf.}}$ | Sig. |
|---|---|---|---|---|
| *Hybrid strategy* vs Random Policy | $6.95 \times 10^{-85}$ | $1.04 \times 10^{-84}$ | $2.08 \times 10^{-84}$ | *** |
| *Hybrid strategy* vs PPO (reward +1/0) | $6.03 \times 10^{-18}$ | $6.03 \times 10^{-18}$ | $1.81 \times 10^{-17}$ | *** |
| *Hybrid strategy* vs QMIX (reward +10/0) | $7.52 \times 10^{-89}$ | $2.26 \times 10^{-88}$ | $2.26 \times 10^{-88}$ | *** |

Table A.10: Pairwise comparisons of Episode Length. Reported $p$-values from Mann–Whitney U tests, adjusted within each family of three tests using Benjamini–Hochberg (BH) and Bonferroni.

| Comparison | $p_{\text{raw}}$ | $p_{\text{BH}}$ | $p_{\text{Bonf.}}$ | Sig. |
|---|---|---|---|---|
| *Hybrid strategy* vs Random Policy | $2.15 \times 10^{-135}$ | $6.45 \times 10^{-135}$ | $6.45 \times 10^{-135}$ | *** |
| *Hybrid strategy* vs PPO (reward +1/0) | $4.97 \times 10^{-84}$ | $7.46 \times 10^{-84}$ | $1.49 \times 10^{-83}$ | *** |
| *Hybrid strategy* vs QMIX (reward +10/0) | $9.80 \times 10^{-56}$ | $9.80 \times 10^{-56}$ | $2.94 \times 10^{-55}$ | *** |

### A.8.2 FOUR AGENTS ENVIRONMENT

Evaluation in a more complex multi-agent setting was conducted in a 16×16 grid-world with four agents and three apple trees (see Figure A.7). In this environment, each tree disappears permanently once all surrounding apples are harvested, introducing a localized resource depletion mechanism and stronger interdependencies among agents. Moreover, resources can only regenerate up to a much lower threshold, as determined by the regeneration rule in Equation 1, where the parameter $\tau$ is fixed at 16. This implies that, regardless of the initial distribution across multiple trees, regrowth cannot exceed the equivalent of a single tree. This design increases both the spatial complexity and the consequences of overexploitation, thereby raising the challenge of sustaining cooperative resilience.

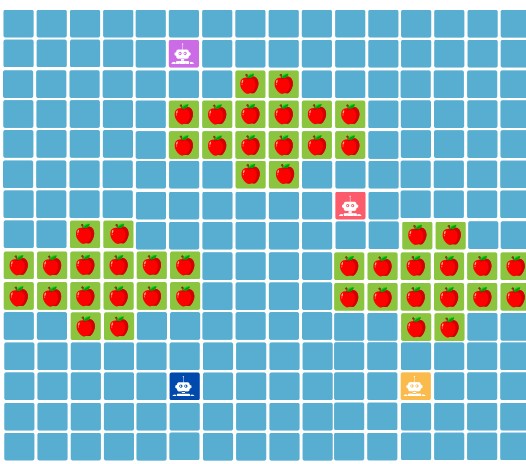

Figure A.7: Extended 16×16 grid-world environment with four agents and three apple trees. Each tree permanently disappears once all surrounding apples are harvested, introducing localized resource depletion and stronger interdependencies among agents.

The results of 50 episodes are summarized in Figure A.8 using boxplots for cooperative resilience metrics, average final agent consumption, episode length, and bar plot for the number of episodes in which the last apple was taken. As shown, the *hybrid strategy* achieves higher values in consumption and episode length, while reducing the frequency of last-apple events associated with social dilemma failures.

To further assess the robustness of our method, we performed Mann–Whitney U tests for each metric, along with Benjamini–Hochberg and Bonferroni corrections. The detailed results are reported in Tables A.11, A.12, and A.13 for cooperative resilience, agent consumption, and episode length, respectively. In the Tables stars indicate significance levels (* $p < 0.05$, ** $p < 0.01$, *** $p < 0.001$). In the Table A.11 the statistical analysis confirms that *hybrid strategy* significantly outperforms both the random policy baseline and standard PPO in terms of cooperative resilience. While the strongest effect is observed against the random baseline ($p < 10^{-9}$), the improvement over PPO also reaches statistical significance ($p < 0.05$ after corrections), indicating that resilience-based rewards provide a measurable advantage in guiding cooperative behavior. The results for average agent consumption, reported in Table A.12, reinforce the previous finding: *hybrid strategy* significantly outperforms both the random baseline and PPO ($p < 0.001$ after all corrections).

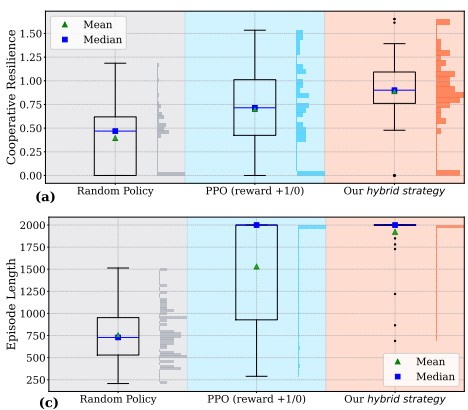
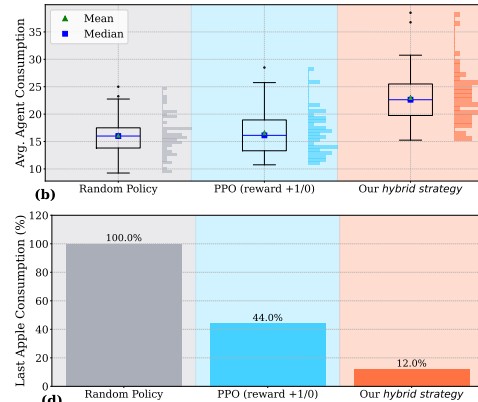

Figure A.8: Performance metrics over 50 episodes in the extended 16×16 environment with four agents and three apple trees. **(a)** Cooperative resilience. **(b)** Average total apple consumption per episode across both agents. **(c)** Episode length. **(d)** Last-apple consumption frequency, indicating the occurrence of social dilemma failures.

Table A.11: Pairwise comparisons of cooperative resilience. Reported $p$-values from Mann-Whitney U tests, adjusted within each family of three tests using Benjamini-Hochberg (BH) and Bonferroni. For 16×16 environment.

| Comparison | $p_{\text{raw}}$ | $p_{\text{BH}}$ | $p_{\text{Bonf.}}$ | Sig. |
|---|---|---|---|---|
| *Hybrid strategy* vs Random Policy | $2.77 \times 10^{-10}$ | $5.53 \times 10^{-10}$ | $5.53 \times 10^{-10}$ | *** |
| *Hybrid strategy* vs PPO (reward +1/0) | $2.27 \times 10^{-2}$ | $2.27 \times 10^{-2}$ | $4.54 \times 10^{-2}$ | * |

Table A.12: Pairwise comparisons of Average Agent Consumption. Reported $p$-values from Mann–Whitney U tests, adjusted within each family of three tests using Benjamini–Hochberg (BH) and Bonferroni. For 16×16 environment.

| Comparison | $p_{\text{raw}}$ | $p_{\text{BH}}$ | $p_{\text{Bonf.}}$ | Sig. |
|---|---|---|---|---|
| *Hybrid strategy* vs Random Policy | $3.20 \times 10^{-11}$ | $6.39 \times 10^{-11}$ | $6.39 \times 10^{-11}$ | *** |
| *Hybrid strategy* vs PPO (reward +1/0) | $4.00 \times 10^{-10}$ | $4.00 \times 10^{-10}$ | $8.00 \times 10^{-10}$ | *** |

Moreover, the results for episode length, shown in Table A.13, confirm that *hybrid strategy* produces significantly longer episodes than both the random baseline and PPO ($p < 0.001$ after corrections). Together with the resilience and consumption results, this indicates that resilience-based rewards lead to more sustainable system dynamics, extending the survivability of the environment while maintaining higher levels of cooperative performance.

Table A.13: Pairwise comparisons of Episode Length. Reported $p$-values from Mann–Whitney U tests, adjusted within each family of three tests using Benjamini–Hochberg (BH) and Bonferroni. For 16×16 environment.

| Comparison | $p_{\text{raw}}$ | $p_{\text{BH}}$ | $p_{\text{Bonf.}}$ | Sig. |
|---|---|---|---|---|
| *Hybrid strategy* vs Random Policy | $4.43 \times 10^{-18}$ | $8.87 \times 10^{-18}$ | $8.87 \times 10^{-18}$ | *** |
| *Hybrid strategy* vs PPO (reward +1/0) | $1.72 \times 10^{-4}$ | $1.72 \times 10^{-4}$ | $3.43 \times 10^{-4}$ | *** |

# B    INTERPRETATION OF LEARNED WEIGHTS

In this section, we analyze the weights learned by the best-performing reward inference configuration, which relied on hand-crafted features, a margin-based preference learning approach, and a

sampling mixture strategy (MPL-M1). This configuration, referred to as *our hybrid strategy* in the main text, achieved the highest cooperative resilience and lowest incidence of selfish behavior, also offers an interpretable structure by design.

Unlike state-based or neural parameterizations, the handcrafted reward function is explicitly defined over a set of six domain-specific features (see Section A.6.1), allowing us to directly trace the contribution of each factor to the learned incentive signal. This transparency provides insights into how the model internalizes resilience, promoting behaviors, and aligns them with measurable aspects of the environment. The learned weights thus act as a diagnostic lens, revealing which characteristics of the system were the most influential in shaping cooperative and sustainable agent behavior.

Table B.1: Interpretation of learned weights for the best handcrafted reward configuration.

| Feature | Description | Learned Weight | Interpretation |
|---------|-------------|:--------------:|----------------|
| $\phi_1$ | **Remaining Apples** (total apples on the grid) | $w_1$ | **Positive weight** $\rightarrow$ agents are incentivized to preserve resources. Encourages sustainability. **Negative weight** $\rightarrow$ preference for depleted environments (possible artifact of training dynamics). |
| $\phi_2$ | **Agent 1 Proximity** (distance to nearest apple) | $w_2$ | **Positive weight** $\rightarrow$ avoidance behavior, potentially to reduce competition. **Negative weight** $\rightarrow$ encourages closer proximity to resources, signaling proactive behavior. |
| $\phi_3$ | **Agent 2 Proximity** (distance to nearest apple) | $w_3$ | Similar to $\phi_2$. If the sign differs from $w_2$, it may suggest role specialization or asymmetric coordination between agents. |
| $\phi_4$ | **Proximity difference** (distance inequality) | $w_4$ | **Positive weight** $\rightarrow$ penalizes asymmetric access to nearby apples. Encourages fairness and spatial balance between agents. **Negative weight** $\rightarrow$ negative weights may signal tolerance for emergent role differentiation. |
| $\phi_5$ | **Local Apple Density (Agent 1)** | $w_5$ | **Positive weight** $\rightarrow$ encourages Agent 1 to operate in resource-rich areas. If significantly larger than $w_6$, may indicate structural bias. |
| $\phi_6$ | **Local Apple Density (Agent 2)** | $w_6$ | Same interpretation as $\phi_5$, but for Agent 2. Comparing $w_5$ and $w_6$ helps diagnose symmetry or potential role division. |

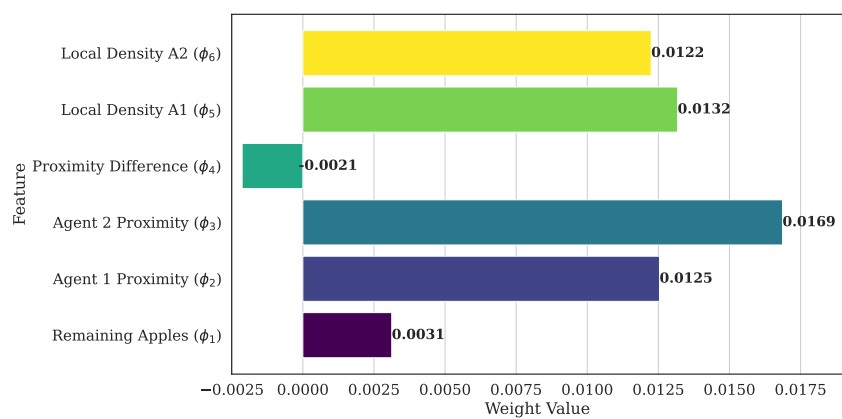

Figure B.1: Learned Weights with the handcrafted reward function in *hybrid strategy*.

Figure B.1 illustrates the learned weights associated with the handcrafted reward function in the best-performing configuration. The relative magnitudes of the weights provide interpretable signals about the emergent behaviors favored by the inferred reward.

Among the six features, the highest weights were assigned to *Agent 2 Proximity* ($\phi_3$), *Local Apple Density for Agent 1* ($\phi_5$), and *Agent 1 Proximity* ($\phi_2$). Since proximity features encode the distance to the nearest apple, their high positive weights indicate that agents were encouraged to maintain a degree of distance from individual resources, possibly to reduce direct competition or avoid over-exploitation. In contrast, the high weights on local density suggest that agents were incentivized to remain within resource-rich areas, without necessarily needing immediate access to a single apple. This combination promotes spatial behaviors in which agents position themselves strategically, away from direct contention, but still within productive regions.

The feature *Remaining Apples* ($\phi_1$) received a smaller positive weight, suggesting that system-level sustainability was modestly rewarded but not heavily prioritized. *Proximity Difference* ($\phi_4$) captures the absolute difference between the distances of both agents to their nearest apple, quantifying potential asymmetries in immediate access to resources. A negative weight on $\phi_4$ might, at first glance, suggest that the model favors *inequality* in access to resources. However, in our setting, this outcome is more plausibly interpreted as the emergence of complementary behaviors, where agents adopt specialized roles. For instance, one agent may consistently position itself closer to resources to prioritize collection, while the other remains farther away, potentially exploring. These spatial asymmetries are not inherently unfair, in fact, they can reflect strategies that enhance overall system performance. A detailed interpretation of each feature and its associated weight is presented in Table B.1.

Interestingly, agents trained under this reward displayed specialized spatial behaviors. One agent actively explored broader areas of the grid, while the other remained anchored along the right boundary, consistently harvesting nearby resources with limited movement. This emergent division of roles can be interpreted as a consequence of the learned incentives, which balance the need to exploit resource-rich zones while avoiding redundant overlap. Such specialization illustrates how the reward structure supports complementary strategies. See supplementary material for a video example of trained agent behavior https://youtu.be/S9UqFlKAgwE.

## C  ADDITIONAL SPATIAL BEHAVIOR VISUALIZATIONS

In this appendix, we provide additional visualizations of agent positions. Figure C.1 shows disaggregated frequency maps for each agent, while Figure C.2 presents circle-based maps where the size of each circle indicates how frequently a location was visited. The latter makes it easy to see the specialized behavior in the *hybrid strategy*, where one agent stays near the boundary and the other explores more widely.

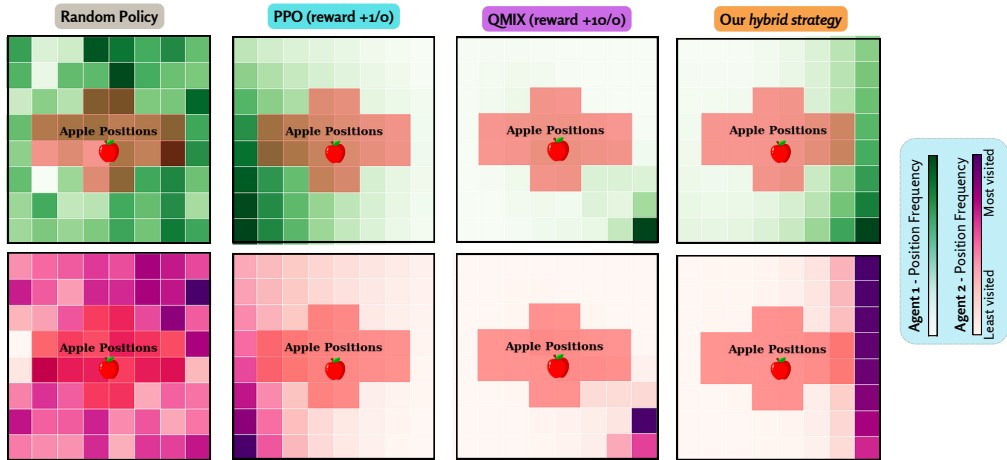

Figure C.1: Individual position frequency maps for Agent 1 (green) and Agent 2 (purple) under four training configurations: (i) random policy, (ii) PPO with standard rewards, (iii) QMIX, and (iv) our *hybrid strategy*. Each heatmap depicts the spatial visitation density over 500 evaluation episodes in an $8 \times 8$ grid, with apple locations marked in red. Agents were randomly initialized at the start of each episode and evaluated under the same protocol with three disruption events.

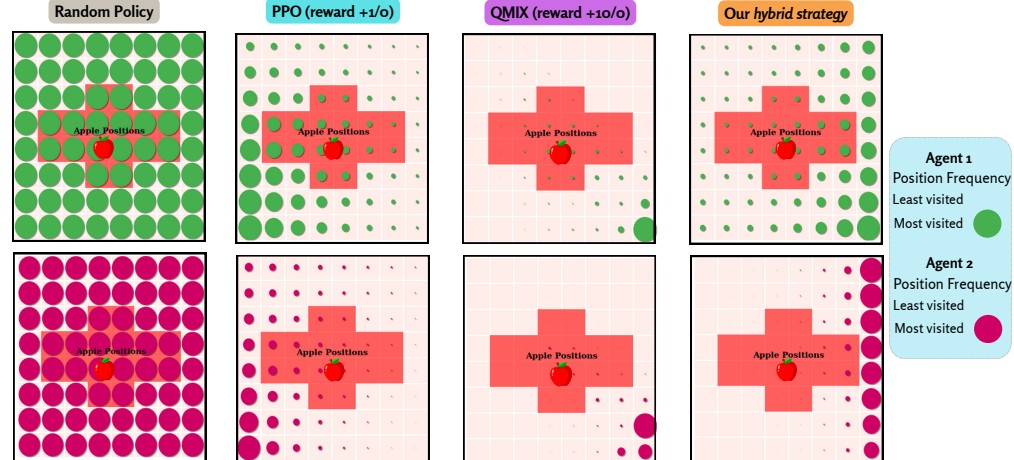

Figure C.2: Circle-based position maps for Agent 1 (green) and Agent 2 (purple) under the four training configurations: (i) random policy, (ii) PPO with standard rewards, (iii) QMIX, and (iv) our *hybrid strategy*. Circle size is proportional to the visit frequency at each grid location, and very low-frequency visits are omitted for clarity.

## D  SOCIETAL IMPACT CONSIDERATIONS

This work contributes to the design multi-agent AI systems that are more resilient to disruptions in shared-resource environments. By focusing on reward learning aligned with collective welfare and recovery, the framework promotes agent behaviors that go beyond individual optimization, encouraging fairness, sustainability, and adaptability.

**Positive Impacts.**   The proposed approach may contribute to the development of multi-agent AI systems capable of operating in dynamic environments such as disaster response, resource management, and multi-robot coordination, where resilience in support of collective well-being is essential. Reward functions inferred from resilient behaviors can act as proxies for socially desirable values, including equity, mutual support, and long-term sustainability—even in systems composed of both artificial and non-artificial (e.g., human) agents.

**Negative Impacts and Limitations**   If misused or deployed without oversight, cooperative reward functions could be manipulated to favor specific agents or reinforce harmful coordination patterns (e.g., excluding certain agents from resources). Furthermore, our method relies on trajectory ranking heuristics that may not fully capture human values. In real-world deployment, this may raise risks related to fairness, representation, or trust.

**Mitigation Measures.**   We encourage researchers to incorporate human-in-the-loop oversight when defining ranking criteria or interpreting resilience-based rewards, especially in domains with ethical, social, or legal implications. Transparency in how reward functions are learned and applied is critical for accountability and alignment with stakeholder values.

While this work does not involve direct deployment or the use of sensitive data, its broader applicability to cooperative AI systems calls for careful consideration of potential misuse—particularly in socio-technical contexts, high-risk domains involving human participants, or settings where agent decisions directly impact real-world resource management.

## E  LIMITATIONS

Our results demonstrate that resilience-aligned rewards improve cooperative behavior in mixed-motive settings. However, the experiments are conducted in a simplified and fully observable environment with discrete actions and known dynamics. We additionally introduced a more complex

setting with a larger search space, more agents, and apple trees with limited regeneration, yet the conditions remain constrained to grid-world environments. Thus, these experiments may not fully capture the complexity of real-world multi-agent systems, where agents often face partial observ-ability, larger and continuous state spaces, and dynamic interaction topologies. Additionally, the resilience metric used to rank trajectories is manually designed for the specific task, which may limit its applicability across domains.

## F  REPRODUCIBILITY ASSETS

In addition to the main manuscript, we provide as supplementary material a GitHub repository containing the source code and experimental configurations used in this study. The repository will be anonymized during the review process and made publicly available upon acceptance. A `README.md` file offers detailed instructions for reproducing the experiments. For completeness, a copy of the source code is also included with this submission as a compressed `.zip` file.

