# OpenReview forum: "Learning Reward Functions for Cooperative Resilience in Multi-Agent Systems"
_ICLR.cc/2026/Conference — Submitted to ICLR 2026_

### Official Review · Reviewer_YJon · 2025-10-19

**Soundness:** 1
**Presentation:** 1
**Contribution:** 2
**Rating:** 2
**Confidence:** 4

**Summary:**

The paper focuses on reward or mechanism design for the mixed-motive game Harvest, where two agents can collect apples with a state- and location-dependent regrowth rate. It proposes a reward learning method based on preference-based inverse reinforcement learning (IRL), consisting of two steps:
1. The experience trajectories are ranked w.r.t. a cooperative resilience metric, which is specifically designed for the Harvest domain, including the number of consumed apples, available apples, inequality of consumption, etc.
2. A reward function is learned based on the ranking using preference-based learning via handcrafted features, linear function, and neural networks, as approximation variants.

The approach is evaluated on small instances of the Harvest domain, consisting of an 8x8 grid and 2 agents (or 16x16 and 4 agents), and compared with baselines, such as a Random policy, PPO, and QMIX.

**Strengths:**

The paper has a clear focus on reward design for mixed-motive games. It is mostly well-written and easy to follow.

The proposed method is interesting and novel.

**Weaknesses:**

**Novelty**

While the proposed method seems novel, the paper misses important discussions (and experimental comparisons) about prior work, which also shapes the behavior of self-interested agents via rewards [1,2,3,4,5].

**Soundness**

There are some flaws in its definition and assumptions:
- The paper formulates the problem as an MDP, despite focusing on multi-agent settings. It is unclear if the MDP refers to a single agent (which would be flawed due to violating the Markov property [6]) or the joint multi-agent view (which would represent a multi-agent MDP or MMDP [7]).
- The paper assumes full observability, contradicting the original setting of Harvest, which is defined as a partially observable stochastic game [8,9].
- The paper assumes information sharing between the agents, e.g., the experience trajectories containing the joint actions, which is unrealistic in mixed-motive scenarios, where **(1)** agents are assumed to be independent [8,9,10], and **(2)** global and perfect communication is required, thus limiting scalability.

The proposed method is specifically designed for the Harvest domain, as the cooperative resilience metric consists of four indicators (the paper states *"five"* but I could not find the fifth one):
1. Cumulative consumption of apples
2. Resource/Apple availability
3. Gini index (distribution of apple consumption)
4. Hunger index (number of time steps without any consumption)

I wonder how these metrics would map to other common social dilemmas, such as Cleanup, Public Goods, Coin, and Wolfpack.

To further strengthen the contribution, a theoretical analysis would have been helpful, e.g., where the concept is shown to work in the iterated prisoner's dilemma.

**Significance**

The introduction of the paper puts a strong emphasis on environmental disruptions. However, throughout the paper and experimental section, I do not find such disruptions that would validate the claims regarding cooperative resilience (the metric is reported in the experiments, but I am unaware of the actual disruptions, if any). I recommend testing variations, e.g., where the regrowth rate of apples is varied [8,9] or communication channels are noisy [5]. Without such an evaluation, I cannot confirm its significance.

The experiments are somewhat preliminary, as they focus on very small instances of the Harvest domain with four agents at most. Prior work evaluates with 8-12 agents in the Harvest domain [1,5,8], as well as other domains, such as Cleanup or Coin [2].

Due to the lack of scaling and variety of test environments, I am concerned about the generality and scalability, and consequently, the broader relevance of the approach.

**Literature**

[1] Lupu et al., "Gifting in Multi-Agent Reinforcement Learning", AAMAS-20

[2] Yang et al., "Learning to Incentivize Other Learning Agents", NeurIPS-20

[3] Schmid et al., "Stochastic Market Games", IJCAI-21

[4] Vinitsky et al., "A learning agent that acquires social norms from public sanctions in decentralized multi-agent settings", Collective Intelligence 2023

[5] Phan et al., "Emergent Cooperation from Mutual Acknowledgment Exchange in Multi-Agent Reinforcement Learning", JAAMAS-24

[6] Littman et al., "Markov Games as a Framework for Multi-Agent Reinforcement Learning", ICML-94

[7] Boutilier et al., "Planning, Learning and Coordination in Multiagent Decision Processes", TARK-96

[8] Perolat et al., "A multi-agent reinforcement learning model of common-pool resource appropriation", NeurIPS-17

[9] Leibo et al., "Multi-agent Reinforcement Learning in Sequential Social Dilemmas", AAMAS-17

[10] Foerster et al., "Learning Opponent-Learning Awareness", AAMAS-18

**Questions:**

Regarding the cooperative resilience metric: Cumulative consumption is potentially unbounded, whereas the Gini index has strict bounds. How does the metric ensure that these indicators are weighted fairly?

---

> ### Author Response · Authors · 2025-11-20
> **General Response Summary**
>
> First, we express our sincere gratitude for the opportunity to revise our manuscript. We greatly appreciate the constructive feedback and insightful suggestions provided, which have helped us improve the quality of the work. In follows comments, we provide detailed responses to each point raised, addressing specific questions and broader concerns about the paper. For ease of reference, we denote reviewer questions by the label Q and more general comments/criticisms by the label C. In the revised version of the manuscript, all changes introduced in response to the reviews will be highlighted in blue, while existing content that was already present in the original submission but is now emphasized for clarity will be highlighted in orange. Each modification will be annotated with a tag of the form Ci or Qi, accompanied by the corresponding reviewer alias, to make the mapping between feedback and revisions.

---

> ### Author Response · Authors · 2025-11-20
> **Q1**
>
> Thank you for this important question. This is indeed an important point when combining indicators with different natural scales.
>
> Our cooperative resilience metric is grounded in the framework established in Chacon-Chamorro et al. (IEEE TAI, 2025) and in the general resilience-measurement formalism of Ayyub (Risk Analysis, 2014). Both works define resilience as a performance–area ratio computed over different phases of degradation and recovery induced by disruptions (e.g. failure, and recovery intervals). Under this formulation, the absolute scale of each underlying indicator, whether bounded or unbounded, does not affect the final range of the resilience score, which is determined by the ratio of performance areas rather than by raw magnitudes.
>
> Although cumulative consumption is unbounded, the resilience score is not a linear aggregation of raw indicators. Instead, for each indicator, we first construct a performance curve over time. Following Ayyub's formulation, we then identify the disturbance windows corresponding failure, and recovery. Within these windows, we compute: i) its area under normal (no-disruption) conditions, over the same failure–recovery intervals, ii) its area under disruption, again over the same time windows, iii) the ratio between these two areas, to which we apply the resilience equation proposed by Ayyub (2014) to obtain a resilience score for that indicator. This quantity is a measure in the mathematical sense: non-negative, scale-invariant with respect to the raw indicator, and comparable across indicators.
>
> Under this construction, the final metric is inherently normalized, independently of whether the underlying signal is bounded (e.g., the Gini index) or unbounded (e.g., cumulative consumption). The ratio lies in [0,1] for typical degradation patterns and may exceed 1 for super-resilient behaviors where system performance under disruption surpasses the baseline.
>
> When multiple disruptions occur in a single episode, we follow Chacon-Chamorro et al. (2025) and compute a time-aggregated resilience profile that penalizes sustained performance drops and rewards recovery and improvement, thereby capturing not only resistance but also transformational adaptation of the system. Finally, to obtain a single cooperative-resilience score, we aggregate the per-indicator resilience values using the harmonic mean, which further enforces fairness: indicators with disproportionately high values cannot dominate the composite metric, while degraded indicators have a stronger impact, consistent with the requirement that a resilience measure should be sensitive to weaknesses rather than to isolated peaks.

---

> ### Author Response · Authors · 2025-11-20
> **C1. [Prior work on reward shaping and incentive design]**
>
> Thank you for raising this point. We agree that mechanisms such as peer rewarding, in which agents can deliberately influence each others' reward function (Lupu et al., 2020) (Yang et al., 2020) norm formation via sanctions (Vinitsky et al., 2023), and mutual acknowledgment (Phan et al., 2024) are central contributions in MARL incentive design. We revise the related-work section to explicitly connect our approach to these lines.
>
> However, the goals and methodological foundations of these works differ fundamentally from our contribution in several ways. First, prior methods primarily aim to shape incentives during training so that self-interested agents learn cooperative behavior. In contrast, our work focuses on learning a reward function directly from ranked trajectories, where the rankings themselves encode cooperative resilience. Our objective is not only to elicit cooperative behavior, but also to ensure that such behavior remains resilient under disruptions, a dimension that is not explicitly modeled in the cited approaches.
>
> In the revised manuscript, we therefore add a subsection explicitly contrasting reward inference from cooperative resilience based rankings (our method) with reward gifting and norm/incentive shaping for cooperation (the cited works). We also clarify that our goal is not to replace incentive-exchange mechanisms, our aim is to introduce a cooperative resilience-grounded reward learning framework that can, in principle, complement or be integrated with them.

---

> ### Author Response · Authors · 2025-11-20
> **C2. [Soundness: MDP formulation, observability, and information sharing]**
>
> Thank you for raising this point; we agree that our exposition in the current version can be clearer, and we revise the manuscript accordingly.
>
> We thank the reviewer for pointing out the ambiguity in our formulation. Our intention was never to model the interaction as a single-agent MDP, but rather to adopt the standard joint-state, joint-action view of a multi-agent system. This corresponds to a multi-agent MDP / Markov game in which the state $s \in S$ is the global environment state and $A$ is the joint action space, consistent with the formulations in [6,7].
>
> To avoid confusion, in the revised manuscript we will explicitly include this. Our reward-learning procedure is then applied in this centralized, joint-process view, while each agent still executes its own policy in the usual decentralized way.
>
> On the other hand, you are correct that the original environment is partially observable. In our current work, we deliberately adopt a fully observable variant, and we will make this modeling choice explicit in the revised manuscript.
>
> Extending our framework to the original partially observable environment is technically feasible but requires additional and  more complex elements: resilience indicators would need to be defined over belief states or observation histories, and the learned reward would be parameterized over those beliefs/histories rather than the full state. We view this as an important and non-trivial next step, and we are currently investigating this direction in follow-up work. The fully observable case studied here is therefore a \textbf{necessary first step that establishes the viability of resilience-based reward inference} before tackling the more challenging partially observable setting.
>
> Finally, we agree that in mixed-motive settings communication or information exchange between agents can be unrealistic. Here, however, our assumptions differ in an important way, and we will clarify this in the revised manuscript. Even though the environment is fully observable, agents do not exchange messages, share intentions, transmit rewards, or reveal learning signals. They simply observe the global state provided by the environment. This is consistent with many fully observable Markov games used in cooperative and mixed-motive MARL, and does not imply any coordination channel beyond environmental observation. Our agents still face conflicting incentives irrespective of whether the environment is globally observable.
>
> Moreover, we acknowledge that, at very large scales, storing full joint trajectories may become costly. In the revised manuscript we explicitly recognize this.

---

> ### Author Response · Authors · 2025-11-20
> **C3. [Soundness: Domain-specificity of the indicators]**
>
> Thank you for raising this point. The reference to “five” indicators was imprecise, and we will correct this in the revised manuscript. In the 2-agent setting, the cumulative consumption indicator is computed per agent (i.e., two separate curves), while availability, inequality (Gini), and hunger are system-level indicators, resulting in five total performance curves. In the 4-agent settings, the same logic applies: consumption curves expand to one per agent, while the remaining indicators are aggregated at the population level. In the revised version, we clarify this.
>
> Regarding domain specificity, we agree that the four indicator types are tailored to Commons Harvest, but their underlying principles: resource availability, fairness, consumption dynamics, and deprivation, are generalize naturally to other social dilemmas. For example: Cleanup can use resource availability like waste removal level; hunger like contribution lapses. Public Goods can use consumption like individual contributions and Gini like contribution inequality. Coin or Wolfpack: availability is related with task opportunities.
>
> We fully agree that the selection of indicators for computing the resilience score is environment-dependent and necessarily involves domain knowledge. In our framework, the goal is provide a generalizable structure: once relevant indicators of collective well-being are selected for a given environment, our methodology enables the construction of a cooperative resilience score and the subsequent inference of reward functions that promote those desired system-level properties.
>
> We emphasize that the selection of indicators is **problem-specific**, and this aligns with practices in other domains such as psychology and critical infrastructure studies, where resilience is assessed via contextually relevant metrics. These fields highlight that resilience is not a single value but a property that emerges from system behaviors in response to disruption, measured along dimensions that reflect what is important for the particular system.
>
> Ultimately, our contribution lies in offering a flexible framework: given an environment and a conceptualization of what constitutes collective well-being, one can define appropriate indicators, use them to compute a resilience-based trajectory ranking, and then learn reward functions that align agent behavior with those goals.
>
> In the revised version, we explicitly discuss this ideas.

---

> ### Author Response · Authors · 2025-11-20
> **C4. [Significance: Environmental disruptions]**
>
> Thank you for raising this point.  The experiments do include explicit environmental and agent-level disruptions, both during trajectory ranking and during the final evaluation:
>
> - Disruption used during trajectory ranking (Section 4.2): At timestep 500, the environment triggers a resource-removal disruption, where apples from the central tree are removed with a fixed probability, ensuring at least one remains so the episode continues.
> - Expanded disruption protocol used for evaluation (Section 4.4): To avoid overfitting to a single disruption type and to assess generalization, evaluation is performed under three temporally separated and qualitatively distinct disruptions, each lasting 5000 steps: i) Resource removal at step 1250; ii) Temporary reduction in apple regrowth rate at step 2500 until 2600; iii) Agent failure, where one agent loses control and moves randomly from steps 3750 to 3900. A timeline summarizing these disruptions is explicitly shown in Appendix A.3 (Figure A.1). We will make these disruption mechanisms more prominent in the main text (introduction and experimental setup).
>
> Regarding the suggestion of noisy communication channels, in our current setting agents do not communicate at all they receive no explicit messages or acknowledgments from one another, so this type of perturbation does not apply directly to our experimental setup. We will clarify the absence of communication channels in the revised manuscript.
>
> In the revised manuscript, we will highlight in orange the disruption elements that were already present in the original submission, to make clear that these aspects were not added post-hoc.

---

> ### Author Response · Authors · 2025-11-20
> **C5. [Scalability]**
>
> Thank you for raising this point. We appreciate this concern and agree that our experimental setup does not yet match the larger-scale evaluations (8–12 agents, multiple domains) reported in prior work. Our study should be understood as a first step toward integrating cooperative resilience into reward design, rather than as a comprehensive benchmark across all standard social-dilemma environments.
>
> In the main experiments, we focus on a 2-agent 8×8 environment to clearly illustrate the full pipeline: resilience-based ranking, preference-based IRL, and hybrid rewards under disruptions. To partially address scalability, we also include an extended 16×16 environment with 4 agents and 3 apple trees, where local resource depletion and a stricter regeneration threshold introduce stronger interdependencies and more complex dynamics (Table 1, Section 4.5). In this setting, our hybrid strategy still improves episode length, cumulative consumption, and cooperative resilience.
>
> We explicitly acknowledge that full scaling to 8–12 agents and to other benchmarks such as Cleanup or Coin remains an important direction for future work.

---

### Official Review · Reviewer_KjM8 · 2025-10-30

**Soundness:** 3
**Presentation:** 2
**Contribution:** 2
**Rating:** 6
**Confidence:** 4

**Summary:**

This paper introduces a preference-based inverse reinforcement learning (IRL) framework aimed at discovering reward functions that promote cooperative resilience in multi-agent systems. The approach first quantifies resilience for full trajectories using indicators such as consumption, resource availability, inequality, and hunger. Trajectories are ranked by these resilience scores, and pairwise preferences are used to train reward models via either a margin-based or probabilistic preference-learning objective. The learned rewards are then used—alone or in combination with standard individual rewards—to train PPO-based multi-agent policies. Experiments on Commons-Harvest–style environments and a larger 16×16 scenario show that hybrid rewards reduce resource depletion and increase sustainability compared with Random, PPO, and QMIX baselines.

**Strengths:**

The paper’s main strengths are its clear and timely problem framing—learning reward functions that encode cooperative resilience rather than handcrafting sustainability terms—coupled with a well-structured pipeline that converts trajectory-level resilience scores into pairwise preferences and trains reward models (margin-based or probabilistic) that plug seamlessly into standard MARL; it explores multiple reward parameterizations (handcrafted, linear, neural) and a practical hybrid objective (individual + resilience) that consistently improves long-horizon outcomes such as reduced depletion and longer sustainable operation; the environment setup and evaluation protocol are described with care, including statistical testing and ablations on preference generation strategies, and the presentation emphasizes reproducibility with implementation details and organized appendices, making the contribution both conceptually meaningful and practically usable by the community.

**Weaknesses:**

1. The resilience metric is manually constructed as a harmonic mean over several indicators with fixed weights and failure/recovery windows. The paper does not study how these design choices affect trajectory rankings or learned rewards. Because the metric defines the training signal, its sensitivity is a critical missing analysis.

2. Rewards are learned from single-shock episodes (one disruption at step 500) but tested on triple-shock long runs. The authors claim generalization to unseen disruptions, yet they never quantify how well ranking-based rewards transfer to new disturbance regimes. This leaves it unclear whether improvements stem from genuine resilience learning or from overfitting to the training scenario.

3. Only Random, PPO, and a lightly tuned QMIX variant are compared. More recent cooperative MARL algorithms(MAPPO,HAPPO,COMA) or better-tuned decomposers(QTRAN,QPLEX,VDN) could provide stronger baselines. Without tuning or wall-clock comparisons, improvements may partly reflect hyper-parameter asymmetry rather than reward-learning advantages.

4. The trajectory pairs come from random policies; possible noise, transitivity violations, or ranking bias are not analyzed. Since both MPL and PPL depend on clean ordinal information, unverified ranking noise could distort the learned reward landscape.

5. The larger-scale 16×16 experiment shows mixed or statistically weak resilience gains, and only 50 evaluation episodes are run. The paper also claims interpretability for handcrafted or linear rewards but does not show what the learned weights actually represent or how they correlate with indicators.

**Questions:**

1. How sensitive are results to indicator selection, normalization, and harmonic aggregation?

2. How does performance change if the disruption frequency or type differs between training and testing?

3. What are the exact tuning budgets and compute times for PPO, QMIX, and the preference-learning stages?

4. How stable are learned rewards when preference noise or inconsistent pair rankings are introduced?

5. In the larger-scale setting, which element—preference model, reward parameterization, or hybridization—drives the observed gains?

---

> ### Author Response · Authors · 2025-11-20
> **General Response Summary**
>
> First, we express our sincere gratitude for the opportunity to revise our manuscript. We greatly appreciate the constructive feedback and insightful suggestions provided, which have helped us improve the quality of the work. In follows comments, we provide detailed responses to each point raised, addressing specific questions and broader concerns about the paper. For ease of reference, we denote reviewer questions by the label Q and more general comments/criticisms by the label C. In the revised version of the manuscript, all changes introduced in response to the reviews will be highlighted in blue, while existing content that was already present in the original submission but is now emphasized for clarity will be highlighted in orange. Each modification will be annotated with a tag of the form Ci or Qi, accompanied by the corresponding reviewer alias, to make the mapping between feedback and revisions.

---

> ### Author Response · Authors · 2025-11-20
> **Q1**
>
> Thank you for this important question. Our resilience metric follows the formal structure used in Ayyub (2014) and in Chacón-Chamorro et al. (2025), where resilience is defined as a ratio of performance areas computed independently for each indicator. This construction provides two forms of inherent normalization: i) scale invariance at the indicator level, since each indicator is normalized by its own reference-area baseline, and ii) aggregation robustness, since the harmonic mean prevents any single indicator from dominating the final score.
>
> That said, we agree that we did not perform an explicit sensitivity analysis over the indicator set, its normalization scheme, or the choice of the harmonic mean. Conceptually, the harmonic aggregation was chosen because it penalizes weaknesses, consistent with resilience theory, but we acknowledge that exploring alternative aggregation strategies (geometric mean, weighted schemes) would provide a deeper understanding of how design choices influence the ranking signal and, consequently, the inferred reward.

---

> ### Author Response · Authors · 2025-11-20
> **Q2**
>
> Thank you for this important question. We explicitly considered the risk that training on trajectories ranked under a single disruption (apple removal at step 500) could induce overfitting to that specific disturbance profile. For this reason, all agents trained with the inferred reward are evaluated under an expanded disruption protocol that differs substantially from the one used to produce the rankings.
>
> The evaluation environment introduces three temporally distributed and qualitatively distinct disruptions, each lasting 5000 steps: i) Resource removal at step 1250, ii) Regrowth-rate reduction beginning at step 2500 until 2600, iii) Agent failure (random uncontrolled movement) from steps 3750 to 3900. Thus, the test-time disturbances differ from training not only in frequency, but also in type, duration, intensity, and timing.
>
> Across these new disturbance regimes, the agents trained with the resilience-based reward continue to show improvements in sustainability (episode length), lower last-apple collapse, and higher collective resilience relative to PPO and Random baselines, although, as expected, the absolute values change depending on the severity and number of shocks. This pattern suggests that what the reward captures is the general degradation–recovery structure encoded by the ranking metric.
>
> We acknowledge that performance naturally varies with disruption frequency and type, and we do not claim invariance to all possible disturbance patterns. However, the empirical results indicate that the inferred reward generalizes beyond the training shock, supporting our central idea that cooperative resilience aligned incentives transfer to qualitatively different disturbance regimes.

---

> ### Author Response · Authors · 2025-11-20
> **Q3**
>
> Thank you for this important question. In the current work we did not track wall-clock times with enough precision to report exact compute hours. However, all methods were trained under comparable and modest computational budgets using a single workstation with a single GPU. Importantly, PPO with the standard reward and PPO with the inferred reward had very similar training times, since the reward computation (whether handcrafted, linear, or NN-based) adds negligible overhead once the reward function has been learned. The main runtime difference is between PPO and QMIX, with QMIX being noticeably slower due to its centralized mixer architecture.
>
> The additional cost introduced by our framework lies in the offline preference-learning pipeline: collecting 500 trajectories, ranking them, and solving the MPL/PPL optimization for 27 configurations. This exploration phase was intentionally exhaustive to understand how different parameterizations behave. In practice, once a suitable configuration is identified, a user would not need to train all 27 variants, and the recurring cost of applying the method reduces essentially to training a single reward model.

---

> ### Author Response · Authors · 2025-11-20
> **Q4**
>
> Thank you for this important question, as it directly concerns the reliability of the supervision signal in our framework. In our specific case, the method requires a sufficiently diverse set of trajectories, including examples with high, medium, and low cooperative resilience, to infer a meaningful reward. In early experiments (not included in the main results), we attempted to learn from PPO-generated trajectories. However, their limited variability led to rewards that did not extrapolate well, highlighting that the quality and the diversity of the ranked set are crucial for successful reward inference.
>
> More broadly, the robustness of preference-based learning to noisy or inconsistent rankings is an active research topic. Preference-based methods such as margin-based ranking tend to exhibit some robustness to label noise because the learning objective aggregates over many pairwise comparisons rather than relying on absolute scalar scores. Nonetheless, their performance can degrade when noise becomes systematic or when preference relations are highly inconsistent (see, e.g., Cheng et al., ``RIME: Robust Preference-based Reinforcement Learning with Noisy Preferences,'' arXiv:2402.17257, 2024).
>
> As a natural extension of our work, we envision a future experimental protocol that systematically explores:
>
> - Varying the proportion of high-quality vs.\ noisy trajectories.
> - Studying the effect of training on low- vs.\ high-variance trajectory sets.
> - Analyzing training under small, medium, and large sets of ranked trajectories.
>
> Due to time constraints, we were unable to conduct these studies during the rebuttal period. However, your question has helped us recognize this as a promising direction for future work that can further solidify and generalize our framework.

---

> ### Author Response · Authors · 2025-11-20
> **Q5**
>
> Thank you for this important question. In the larger-scale 16$\times$16, 4-agent setting we did not rerun the full set of 27 configurations, as this would have been so expensive computationally; even the evaluation phase in this environment is considerably expensive computationally than in the 8×8 case. Instead, we directly transferred the best-performing configuration identified in the smaller setting, namely, the Handcrafted MPL-M1 Hybrid model (margin-based preference learning with mixture sampling of random and margin 1).
>
> As a consequence, the current experiments in the larger-scale environment do not allow us to cleanly disentangle the relative contribution of i) the preference model (MPL vs.\ PPL), ii) the reward parameterization (handcrafted vs.\ linear vs.\ NN), and iii) the hybridization scheme. The observed gains in the larger-scale setting should therefore be interpreted as evidence that a resilience-aligned reward, learned with MPL-M1 and handcrafted features, can successfully transfer to a more complex environment, so it is not a full ablation of which component is most critical, as we partially explored in the smaller setting. This is nonetheless encouraging: it suggests a practical workflow in which ablation studies and configuration searches can be carried out in simpler environments and then transferred to more demanding scenarios to assess scalability.
>
> We explicitly highlight this ideas and experimental choice in the revised manuscript.

---

> ### Author Response · Authors · 2025-11-20
> **C1. [Choice of baselines MARL]**
>
> Thank you for raising this point. We agree that comparing against a broader set of cooperative MARL algorithms (e.g., MAPPO, HAPPO, COMA, QTRAN, QPLEX, VDN) would strengthen the empirical section. In this first study, however, our focus is not on benchmarking policy-learning algorithms, but on assessing whether a cooperative resilience aligned reward inferred from trajectory rankings can improve behavior when plugged into standard methods. For that reason, we selected PPO and QMIX as widely used, representative baselines and trained them under a limited and comparable tuning budget.
>
> More advanced methods would indeed be valuable additions, but they would shift the paper toward a large-scale empirical benchmarking effort, potentially obscuring the conceptual contribution of resilience-based reward inference. Moreover, our primary goal is to \textbf{highlight the benefit of the inferred reward relative to traditional rewards}, we do not to claim superiority of any particular policy-learning algorithm. A fair comparison would therefore require, for each MARL method, evaluating both the standard reward and our resilience based reward and measuring the improvement induced by the reward, not by the choice of learning algorithm itself.

---

> ### Author Response · Authors · 2025-11-20
> **C2. [The 16x16 results]**
>
> Thank you for raising this point. We agree that the gains in the 16$\times$16 setting are statistically similar to the PPO baseline. However, the consumption and episode-length metrics still indicate favorable behavior: they show that the resilience construction is reflected in the underlying task performance and system persistence. In this experiment, running 500 evaluation episodes, as we did in the small scale case, would have been computationally expensive, so we opted for 50 episodes as a compromise between statistical reliability and feasibility. The results in this regime should therefore be interpreted as exploratory evidence of transfer: they show that a resilience-aligned reward learned in a simpler environment can still yield meaningful improvements in a more demanding setting.

---

> > ### Comment · Reviewer_KjM8 · 2025-11-25
> >
> > I thank the authors for their response. However I find my experimental suggestions are not fully considered:
> > 1. Regarding Q1, considering the citation, it would be good to show that in plain text(Ayyub (2014) and in Chacón-Chamorro et al. (2025)). Also, no further explicit sensitivity analysis are performed.
> > 2. Regarding Q3, 'modest computational budgets' and 'very similar training times' are not precise metric for measuring the training budget. I suggest the authors perform more comprehensive studies on this.
> > 3. Regarding Q4, I thank the authors to take my advice, further experiments on various trajectories are necessary.
> >
> > Therefore, I suggest the authors to add the additional experiments as suggested. I'll maintain my score.

---

> > > ### Author Response · Authors · 2025-12-03
> > >
> > > We sincerely thank for actively engaging during the discussion phase. We fully agree that expanded sensitivity analyses, more precise training budget quantification, and broader trajectory evaluations would further enrich the study. Unfortunately, given the limited discussion window and computational constraints, it was not feasible to execute new large-scale experiments in this revision cycle. Nevertheless, your suggestions point toward a direction for future extensions of this work, and we are committed to incorporating them in a subsequent version. We are grateful for your constructive input and for the thoughtful evaluation of our submission.

---

### Official Review · Reviewer_FUej · 2025-10-31

**Soundness:** 3
**Presentation:** 3
**Contribution:** 3
**Rating:** 4
**Confidence:** 3

**Summary:**

This paper proposes a framework a model-free approach to promoting group resilience by learning reward functions that encourage cooperative resilience in multi-agent systems facing disruptions. The authors define a cooperative resilience metric that combines several system-level indicators (resource availability, equality, hunger, and sustainability) and use it to rank trajectories. These rankings are then used in a preference-based inverse reinforcement learning setup (both margin-based and probabilistic) to infer a reward function that favors resilient behavior. Agents trained with this learned reward are shown to sustain shared resources and recover more effectively after disruptions in a Commons-Harvest gridworld. The paper argues that this method generalizes beyond the specific environment and can be combined with any MARL algorithm to automatically shape cooperative incentives.


The paper’s motivation is strong and the conceptual framing is original, but the evidence provided doesn’t convincingly support the central claim that the system learns resilience rather than merely overfitting a predefined metric. The use of the same resilience score for both supervision and evaluation introduces a causal ambiguity that seriously limits interpretability. Combined with noisy ranking data from random trajectories, narrow metrics, limited environment diversity, and unorthodox PPO configurations, the current results feel preliminary. The hybrid reward setup also weakens the narrative that this approach automates reward design. For these reasons, I would lean to reject. The idea is promising, but it needs stronger experimental grounding, more rigorous validation, and better evidence that the learned reward captures generalizable, causal resilience.

**Strengths:**

- Clear and coherent methodology. The proposed pipeline of ranking trajectories, learning preferences, and inferring rewards is logically structured and mathematically sound.
- Novel IRL formulation for resilience.
- Reproducibility. The paper provides detailed appendices, configurations, and discusses reproducibility assets and ethical considerations, which is commendable.
- Practical potential. The idea of learning system-level incentives from ranked behaviors could, in principle, be applied in many cooperative domains where manual reward design is difficult.

**Weaknesses:**

- Metric-evaluation circularity. The same cooperative-resilience metric used for ranking trajectories is also used to evaluate success. This makes it impossible to tell whether agents actually learned to be resilient or simply optimized the evaluator. The fact that disruptions occur at the same fixed timestep in both training and testing further amplifies this problem.

- Uninformative supervision data. Rankings are generated from random-policy trajectories, which are likely dominated by stochastic noise rather than meaningful cooperation. There is no noise or variance analysis to show that the ranking signal is informative.

- Narrow evaluation focus. Despite defining multiple indicators, the main evaluation metric is “last-apple consumption,” which reflects only sustainability. Other aspects of resilience, like recovery, fairness, and stability, are buried in the appendix.

- Limited generalization and overstated claims. The experiments are confined to a two-agent discrete gridworld with PPO as the sole training algorithm. The “method-agnostic” claim is not empirically demonstrated.

- Missing related work on group resilience.  Shraga et al. Collaboration Promotes Group Resilience in Multi-Agent RL. - RLC 2025.

**Questions:**

- How do you justify metric–evaluation coupling?

- What were the variance results over resilience scores across seeds.

- How does this work relate to Shraga et al 2025 ?

---

> ### Author Response · Authors · 2025-11-20
> **General Response Summary**
>
> First, we express our sincere gratitude for the opportunity to revise our manuscript. We greatly appreciate the constructive feedback and insightful suggestions provided, which have helped us improve the quality of the work. In follows comments, we provide detailed responses to each point raised, addressing specific questions and broader concerns about the paper. For ease of reference, we denote reviewer questions by the label Q and more general comments/criticisms by the label C. In the revised version of the manuscript, all changes introduced in response to the reviews will be highlighted in blue, while existing content that was already present in the original submission but is now emphasized for clarity will be highlighted in orange. Each modification will be annotated with a tag of the form Ci or Qi, accompanied by the corresponding reviewer alias, to make the mapping between feedback and revisions.

---

> ### Author Response · Authors · 2025-11-20
> **Q1**
>
> Thank you for this important question. In our setting, metric–evaluation coupling is expected rather than problematic, because the goal of the IRL pipeline is specifically to recover a reward function that reproduces and extrapolates the cooperative resilience ordering defined by the metric. In particular, our approach follows the spirit of preference-based IRL methods such as Brown et al. (“Extrapolating Beyond Suboptimal Demonstrations via Inverse Reinforcement Learning”), where the learned reward is trained to generalize the ranking signal beyond the specific trajectories used to construct it. This mirrors standard practice in preference-based IRL, where the same preference signal defines both the supervision and the behavioral objective.
>
> However, several factors prevent circularity and ensure that the learned policy is not merely overfitting the evaluator.**The agent never receives the resilience metric during training**. What it receives is an inferred reward, learned from trajectory-level comparisons, not the metric itself. This reward is a function of the state, not of the trajectory-level resilience score.
>
> Moreover, trajectory rankings are generated using a single, fixed disruption type (apple removal at step 500) in 1000 steps, while evaluation uses the expanded protocol with a different protocol which introduces three temporally distributed and qualitatively distinct disruptions, in 5000 steps: i) resource removal at step 1250, ii) a temporary reduction in apple regrowth rate starting at step 2500 until 2600, and iii) an agent failure simulation, where one agent loses control and moves randomly from steps 3750 to 3900.
>
> This breaks any possible overfitting to a single disruption pattern and shows that the inferred reward induces policies that generalize to unseen disruption types, timings, and durations.
>
> On the other hand, the ranking signal aggregates indicators with distinct temporal dynamics (consumption, fairness/Gini, availability, hunger). Agents cannot trivially “hack’’ the evaluator, because no single behavior improves all indicators simultaneously: achieving high cooperative resilience requires a specialized behavioral pattern in which agents, guided by the inferred reward, must exploit resources without collapsing the environment, recover after shocks, maintain fairness, and avoid last-apple collapse.
>
> In addition, although cooperative resilience is computed from these indicators, we also report separate qualitative behavioral metrics: average consumption (task performance), last-apple frequency (selfish collapse), episode length (sustainability), and spatial distributions of agents under disruption. Crucially, these qualitative metrics are not used in the construction of the resilience score, and thus provide an independent check that the learned reward induces genuinely cooperative resilient behavior not merely optimizing a single scalar evaluator.
>
> We clarify these points in the revised manuscript to avoid giving the impression of circularity.

---

> ### Author Response · Authors · 2025-11-20
> **Q2**
>
> Thank you for this important question. In the current work, we do not perform a systematic multi-seed analysis over training runs. For each configuration, we learn a single reward function and a single policy and then report mean and standard deviation of resilience and other metrics over 500 evaluation episodes (and 50 in the extended 16×16 setting), which captures variability across environments rather than across random seeds. A full multi-seed study would require repeating the entire pipeline (trajectory collection, reward inference, and policy training) for each configuration, which is computationally expensive given the number of configurations already explored. We agree that a multi-seed robustness analysis would strengthen the empirical section, and we acknowledge this as an important direction for future work. Importantly, PPO and QMIX baselines were trained under the same single-seed regime, so all comparisons in the paper are methodologically consistent.

---

> ### Author Response · Authors · 2025-11-20
> **Q3**
>
> Thank you for this important question. Shraga et al. propose group resilience and show that collaboration protocols (communication, message sharing) improve agents’ ability to recover after perturbations. Their focus is therefore on how to train agents to be more resilient by equipping them with collaboration mechanisms.
>
> Our contribution is fundamentally different in scope and intention. We aim to infer a reward function that captures resilience itself from behavioral evidence. In our framework, cooperative resilience is not induced through communication protocols or architectural design, but emerges from the reward inferred via preference-based IRL applied to trajectories ranked with a resilience metric. The learned reward can then be used by any downstream learning algorithm to produce policies with resilience properties.
>
> This distinction matters for generalization. Collaboration-based approaches (including Shraga et al.) promote resilience by adding capabilities, while our approach seeks to uncover the underlying incentive structure that gives rise to resilient behavior, closer to problems like ecological systems, collective biological agents, or human groups where the reward is preferred to be inferred than engineered. Our method is therefore complementary: group-resilient behaviors in works like Shraga et al. could themselves be used as demonstrations from which our IRL approach can extract resilience-aligned rewards.
>
> We include this clarification and cite Shraga et al. appropriately in the revised version.

---

> ### Author Response · Authors · 2025-11-20
> **C1. [Informativeness of the supervision signal]**
>
> Thank you for raising this point. We agree that trajectories generated by a random policy are not “cooperative” in the usual sense. However, in our setting the supervision does not come from the policy itself, but from the cooperative-resilience metric applied to those trajectories under disruption. Even with random behavior, the disruption protocol induces a wide spectrum of resilience outcomes (from early collapse to partial recovery), and we explicitly verified that the resulting scores span a broad range; this distribution is illustrated in the boxplot of resilience values reported in the appendix.
>
> From a theoretical standpoint, preference-based IRL does not require optimal or even highly skilled demonstrations, but rather a consistent ranking signal over trajectories. Following Brown et al. (“Extrapolating Beyond Suboptimal Demonstrations via Inverse Reinforcement Learning”), our method leverages these resilience  rankings to extrapolate a reward that favors trajectories with better degradation–recovery profiles. In other words, the random policy is only a generator of diverse behaviors; the information comes from how the resilience metric organizes those behaviors.
>
> We acknowledge that we did not include an explicit noise/variance analysis of the ranking signal itself, and we note this as an interesting direction for future work. In particular, studying the robustness of the inferred reward under perturbations of the ranking, subsampling of trajectories, or variations in the quality and representativeness of the trajectory dataset would provide valuable insights into how sensitive cooperative resilience based reward inference is to the supervision source.

---

> ### Author Response · Authors · 2025-11-20
> **C2. [Narrow evaluation focus]**
>
> Thank you for raising this point. We agree that “last-apple consumption” alone would be insufficient to characterize resilience, and we did not intend it to be the sole evaluation metric. We highlight it in the main text because it is an intuitive proxy for egoistic collapse and lack of sustainability, but our evaluation is broader because the cooperative-resilience score combines consumption, resource availability, fairness (Gini), and hunger, so whenever we report resilience we are implicitly evaluating recovery, fairness, and stability jointly, not only sustainability. Beyond resilience and last-apple %, we analyze episode length, as a measure of sustainability and system persistence, cumulative consumption, as a measure of task efficiency and spatial distributions of agents under disruptions, as a qualitative spatial indicator of coordination patterns.
>
> On the other hand, in the revised manuscript, we have added a clearer explanation of why the selected configuration is considered “best”, emphasizing that the choice is multi-criteria (resilience, cumulative reward, last-apple consumption, variance) and not based on last-apple consumption alone, which was under-explained in the original submission.

---

> ### Author Response · Authors · 2025-11-20
> **C3. [“method-agnostic” claim]**
>
> Thank you for raising this point. We agree that, in its current form, the empirical section does not support a strong “method-agnostic” claim. In the revision, we soften this wording to emphasize that our framework is in principle compatible with different policy-learning algorithms (since it outputs a reward function that can be plugged into any RL method), but we will no longer present this as an empirically demonstrated property.
>
> Regarding scalability, the core experiments are indeed conducted in a two-agent 8$\times$8 gridworld, but we also include an extended 4-agent, 16$\times$16 scenario with three apple trees and a stricter regeneration threshold, which introduces more demanding coordination and resource-coupling dynamics. In this setting, the hybrid strategy trained with the inferred reward still improves resilience and task performance over PPO and random baselines. We acknowledge that scaling to larger populations and additional domains (e.g., Cleanup, Coin) remains an important direction for future work, we inform this in the main document.

---

### Official Review · Reviewer_Wodu · 2025-11-03

**Soundness:** 3
**Presentation:** 3
**Contribution:** 3
**Rating:** 4
**Confidence:** 3

**Summary:**

The paper tackles the significant challenge of designing reward functions for Multi-Agent Reinforcement Learning (MARL) that foster cooperative resilience in mixed-motive environments subject to disruption. Traditional reward structures often fail here, promoting individual gains at the expense of collective system persistence. The authors introduce a framework leveraging Inverse Reinforcement Learning (IRL) to infer a collective reward component directly from trajectories ranked by a cooperative resilience metric.
The methodology employs preference-based IRL (MPL and PPL) to learn rewards parameterized by handcrafted features, linear models, or neural networks. The central experimental result confirms that a hybrid reward strategy—which balances the learned resilience reward with standard individual consumption incentives—significantly improves robustness. Tested in a Commons Harvest social dilemma under a comprehensive, generalized disruption protocol, the hybrid strategy achieved higher cooperative resilience, extended system sustainability, and drastically mitigated catastrophic resource depletion events (last-apple consumption dropped to 13.2% in testing, compared to over 60% for baselines). The framework is presented as a general, method-agnostic approach to reward design that complements existing MARL algorithms.

**Strengths:**

- Originality

The primary original contribution is the robust methodology for grounding reward inference in a quantitative, system-level metric of cooperative resilience. While Inverse Reinforcement Learning (IRL) is not new, using preference-based IRL (MPL/PPL) derived from trajectories ranked explicitly by their recovery and failure profiles under stress is a novel application pathway for incentive design in MARL. This approach circumvents the conventional IRL dependence on near-optimal expert demonstrations by utilizing quantitative rankings derived from a resilience score, which captures the complex dynamic, temporal, and distributed nature of recovery under disruption. The paper successfully operationalizes the abstract concept of resilience (anticipating, resisting, recovering, and transforming) into a practical learning signal. Crucially, the introduction of the hybrid strategy demonstrates an important, original insight into practical reward alignment in mixed-motive settings. The result is an emergent reward function that leads to specialized, non-overlapping spatial behaviors (one agent exploring, one anchoring/harvesting) that maximize joint welfare, illustrating the power of this incentive structure beyond simple aggregated consumption maximization. This framework provides a structured means to inject long-term persistence goals into existing MARL systems.


- Quality

The authors tested 27 configurations across two learning algorithms, various sampling strategies, and three reward parameterizations. Critically, the reward inference used trajectories generated under a single resource removal disruption, while the final evaluation used a complex protocol featuring three distinct, temporally separated disruptions: resource removal, regrowth rate reduction, and agent failure simulation. This methodology provides strong evidence that the learned rewards result in generalized robustness rather than overfitting to a single failure mode present in the training data. The results are quantified using statistical tests (Mann–Whitney U test with Benjamini–Hochberg correction), confirming that the hybrid strategy significantly outperforms Random, standard PPO, and QMIX in terms of cumulative consumption and episode length. Furthermore, the paper includes a non-trivial scalability test in a larger 16x16 environment with four agents and permanent resource depletion, confirming the core benefits persist in increased complexity. The transparency provided by interpreting the learned weights for the best-performing handcrafted model (Section B) significantly enhances the quality of the analysis, offering clear causality between incentive structure and emergent cooperative behaviors (e.g., incentivizing proximity distance while rewarding local density).


- Clarity

The technical presentation is straightforward and professional. The problem formulation, the definition of the MDP, and the two-step methodology (ranking trajectories, then learning the reward via optimization/probabilistic modeling) are clearly laid out. The mathematical structure defining the resilience score (using failure and recovery profiles via integrals and the harmonic mean aggregation) is detailed enough to follow the underlying mechanism.


- Significance

The paper addresses a fundamental limitation in MARL: how to automatically design incentives for long-term collective welfare under uncertainty. The success of the hybrid strategy demonstrates that resilience and individual productivity are not necessarily a zero-sum game; the method simultaneously achieves the highest average consumption and the lowest social dilemma failure rate (13.2% last-apple consumption) compared to baselines. This is a powerful proof of concept for applications in domains like environmental resource management or decentralized infrastructure control.

**Weaknesses:**

- Quality

The experimental quality suffers from weaknesses in the baseline selection and the dependence on parameterization. The paper compares performance against basic PPO and QMIX and explicitly notes the omission of more recent, high-performing cooperative MARL algorithms. This leaves a significant open question regarding the necessity of the complex two-stage IRL process compared to simpler, modern reward shaping or decentralized planning techniques. Furthermore, the QMIX baseline required a 10x manual reward increase (+10 instead of +1) just to prevent agents from converging to suboptimal areas, suggesting the baseline policies were highly fragile and perhaps not optimally representative of competitive cooperative MARL standards. The overwhelming success of the Handcrafted parameterization compared to the Linear and Neural Network models (e.g., Last Apple % of 1.75% for MPL-M1 Handcrafted vs. 8.75% for MPL-K1 NN Hybrid, Table A.5 vs A.7) suggests a severe limitation in the generalization capacity of the learned models when forced to work from raw state inputs. This heavily implies that the quality of the system’s performance is primarily driven by the expert’s choice of six input features, rather than the ability of the preference learning pipeline to automatically identify resilience-aligned signals in complex, non-linear state spaces.

- Clarity

the paper does not fully clarify the mechanisms behind the selection of the best performing configuration (Handcrafted MPL-M1 Hybrid) over other high-performing variants, particularly the PPL models which often yielded comparable resilience scores with drastically different average rewards (e.g., Table A.2). This suggests that the nuanced trade-offs captured by margin definition and sampling strategy were not fully analyzed or clearly presented.

**Questions:**

1. The resilience metric calculation relies on defining the time of worst degradation ($t_f$) and the recovery endpoint ($t_r$). Given that five system indicators are tracked (e.g., consumption, Gini index, resource availability), which specific indicator defined $t_f$ and $t_r$ in practice, or was a rule established based on the harmonic mean score itself to estimate these critical integration points for trajectory ranking?

2. The Handcrafted model was significantly more successful than the Linear and Neural Network parameterizations in achieving optimal resilience and low selfishness (1.75% last-apple consumption for Handcrafted MPL-M1 Hybrid). Does this heavy reliance on six expert-designed features mean the method fundamentally requires high-quality domain expertise, or is it expected that non-linear models would outperform the handcrafted features if given more training data or different architectures?

---

> ### Author Response · Authors · 2025-11-20
> **General Response Summary**
>
> First, we express our sincere gratitude for the opportunity to revise our manuscript. We greatly appreciate the constructive feedback and insightful suggestions provided, which have helped us improve the quality of the work. In follows comments, we provide detailed responses to each point raised, addressing specific questions and broader concerns about the paper. For ease of reference, we denote reviewer questions by the label Q and more general comments/criticisms by the label C. In the revised version of the manuscript, all changes introduced in response to the reviews will be highlighted in blue, while existing content that was already present in the original submission but is now emphasized for clarity will be highlighted in orange. Each modification will be annotated with a tag of the form Ci or Qi, accompanied by the corresponding reviewer alias, to make the mapping between feedback and revisions.

---

> ### Author Response · Authors · 2025-11-20
> **Q1**
>
> Thank you for this important question. Identifying $t_f$ and $t_r$ is indeed central to the computation of resilience, and we appreciate the opportunity to clarify the procedure. Your comment also shows a very careful reading of the methodology, which we sincerely appreciate.
>
> You are correct that $t_f$ and $t_r$ are the critical integration points that define the temporal windows over which the performance-area ratios are computed. In our implementation, these times are not shared across indicators; instead, they are **computed independently** for each indicator, for the following reasons and with the following rules:
>
> - All indicators experience the disruption at the same onset time $t_d$ but the maximum degradation may occur at different moments depending on the nature of each signal. For example, a resource-removal disruption immediately affects availability, but its effect on Gini or hunger may take longer to manifest. In contrast, an agent-failure disruption may first degrade individual consumption, then inequality, and only later resource availability.
>
> - Since all indicators are defined in the ``higher is better'' direction, we consistently define: $t_f = \arg \min_{t_i \leq t \leq t_r} I_k(t),$ for each indicator $I_k(t)$. This identifies the moment of worst functional degradation for that specific performance dimension.
>
> - Note that full recovery is not always guaranteed, some systems may never return to pre-disruption levels. To handle this consistently For a single disruption, $t_r$ is simply taken as the end of the horizon, since the system either recovers or stabilizes into its post disruption time. For trajectories with multiple disruptions, $t_r$ for indicator $I_k$ is defined as the last timestep before the next disruption event, ensuring that each event has its own integration window.
>
> - Independent $t_f$ and $t_r$ per indicator are needed because this allows the resilience measure to capture different degradation and recovery dynamics across signals. Accordingly, for each indicator, resilience is computed as the area ratio between the disrupted and baseline performance curves over the indicator-specific time window $[t_d, t_f]$ and $[t_f, t_r]$ following the formulation of Ayyub (2014).
>
> - As in Chacón-Chamorro et al. (2025), when multiple disruptions occur, the indicator resilience values are accumulated into a time-aggregated resilience profile, which rewards recovery and penalizes sustained degradation, capturing  resistance and transformation. Finally, we aggregate indicators using the harmonic mean, which prevents any single indicator with a high value from dominating and ensures sensitivity to weaknesses.
>
> We include a clearer description of this procedure in the revised manuscript to avoid any ambiguity.

---

> ### Author Response · Authors · 2025-11-20
> **Q2**
>
> Thank you for this important question. We agree that the strong performance of the Handcrafted parameterization reflects, in part, the fact that these features encode meaningful prior structure about the domain. Using expert-designed features over the state naturally introduces a strong prior, which helps the model identify resilience-relevant patterns even with limited data. In this sense, the Handcrafted variant is indeed the least general and the most dependent on domain expertise.
>
> However, two important points mitigate this concern. i) The fact that our preference-based IRL procedure learns appropriate weights for these handcrafted features, leading to markedly improved resilience and reduced selfishness, indicates that the framework effectively extracts the cooperative resilience signal embedded in the trajectory rankings. Without the ranking-based supervision, even high-quality features would not necessarily yield cooperative or resilient policies. ii) We belive that the underperformance of the Linear and NN models is primarily due to data limitations, not structural flaws, Both parametric families operate over the full joint state and therefore require substantially more trajectory data to learn a reward that captures the non-linear dependencies among resource levels, spatial configurations, and agent interactions. Our dataset (500 ranked trajectories) is relatively small for learning a reward function over a high-dimensional, spatially structured environment. With richer data, more extensive architectural exploration, and targeted regularization, we expect nonlinear models to surpass the handcrafted version.
>
> We acknowledge this in the revised manuscript. Importantly, this study provides valuable evidence that learning a cooperative resilience reward is feasible, and it motivates deeper exploration of neural reward parameterizations, particularly we are extending the framework to partially observable settings, where structured representations and neural encoders will likely be essential.

---

> ### Author Response · Authors · 2025-11-20
> **C1. [Quality]**
>
> Thank you for raising this point.
>
> We acknowledge that the baseline suite does not include the full range of high-performing cooperative MARL algorithms. Our goal in this first paper was not to perform a comprehensive benchmark of MARL methods, but to answer the conceptual question:
>
> *“Can cooperative resilience be used as a signal to learn reward functions that induce resilient collective behavior under disruptions?”*
>
> For this purpose, we selected PPO and QMIX for reasons of stability, compatibility with global or hybrid reward structures, and availability of robust existing implementations.
>
> More advanced methods would indeed be valuable additions, but they would shift the paper toward a large-scale empirical benchmarking effort, potentially obscuring the conceptual contribution of resilience-based reward inference. Moreover, our primary goal is to **highlight the benefit of the inferred reward relative to traditional rewards**, we do not to claim superiority of any particular policy-learning algorithm. A fair comparison would therefore require, for each MARL method, evaluating both the standard reward and our resilience-based reward and measuring the improvement induced by the reward, not by the choice of learning algorithm itself.
>
> On the other hand, we fully agree with the reviewer that Handcrafted features introduce strong domain priors. Their superior performance stems precisely from this: the features encode interpretable, resilience-relevant structure about the environment (distance to apples, local densities, joint availability patterns).
>
> However, this does not imply that the IRL pipeline is weak. As discussed above, the method does correctly infer the weights for given meaningful features. Without our framework, the same features would not automatically yield resilient or cooperative behavior. The underperformance of the raw-state models is mainly due to data scarcity and architectural simplicity: the NN models were intentionally kept lightweight and trained on only 500 ranked trajectories, which is modest relative to the complexity of the full 8×8 state. With richer data and more expressive architectures (e.g., CNNs, spatial encoders, attention-based models), we expect nonlinear parameterizations to surpass the handcrafted variant, and we will explicitly highlight this as an important direction for future work.
>
> Finally, the reviewer raises a crucial conceptual question: Why use a complex pipeline instead of simpler shaping or decentralized planning? Our answer is that none of the shaping based MARL methods explicitly optimize for resilience, less for cooperative resilience, they optimize for cooperation in static conditions, not cooperation that recovers from or adapts to disruptions. In this sense, our two-stage pipeline is necessary because cooperative resilience is a trajectory-level structural property, it must be inferred from performance degradation and recovery patterns, and it cannot be simply encoded by giving +r or -r shaping bonuses.  Thus, even if more sophisticated MARL baselines are eventually added, the conceptual value of cooperative resilience based reward inference remains distinct.

---

> ### Author Response · Authors · 2025-11-20
> **C2. [Clarity]**
>
> Thank you for raising this point. We agree that the manuscript did not clearly articulate the criteria used to select the “best-performing” configuration. We clarify this in the revision.
>
> While several PPL variants achieve very high resilience values (e.g., PPL-R and PPL-K in Table A.2), their performance profiles have low average rewards, this indicates that PPL solutions achieve high resilience by inducing overly conservative policies that preserve resources but affects the individual task. Such behavior is resilient but not desirable.
>
> Our selection criterion was therefore multi-objective, not resilience alone. The “best” configuration was the one that simultaneously achieved: high resilience, high rewards, low last-apple consumption, and low variance. Under this multi-criteria view, MPL-M1 Hybrid dominated other variants. We appreciate your comment and will make the selection rationale explicit in the revised text, including a short discussion in the appendix comparing MPL and PPL trade-offs in terms of margin definition and sampling strategy.

---

### Author Response · Authors · 2025-12-03
**Author Final Remarks**

We sincerely thank the reviewers for their constructive comments and for highlighting important points that helped us significantly strengthen the manuscript. Below we summarize the key clarifications and improvements that arose from the rebuttal and revision process.

- **A novel perspective for MARL reward design**: Our work introduces a reward-inference framework that extracts cooperative resilience directly from behavioral trajectories under disruptions, a perspective that, to our knowledge, is absent from current MARL reward-design paradigms. This contribution is fundamentally different from sustainability shaping, norm-learning, or communication based coordination. Multiple reviewers explicitly acknowledged the relevance and clarity of this framing.

- **Full clarification of technical concerns**: All questions regarding the MDP structure, indicator scaling, observability assumptions, and metric aggregation were fully addressed. The revised version now: ii) clearly specifies the multi-agent MDP, global information flow, and observability assumptions; ii) removes any ambiguity about the two-stage learning/evaluation protocol; iii) highlights our mismatched disruption protocol evaluation uses disturbance patterns that differ qualitatively from those seen in training, ensuring that resilience aligned behavior is not an artifact of overfitting to a specific disruption sequence. These clarifications make the methodological pipeline transparent and reproducible.
- **Comprehensive responses**: Although reviewer participation in the discussion phase was limited, we addressed every single point with precise technical clarifications and corresponding manuscript updates. All requested explanations, metric behavior, baselines, scaling, indicator selection, domain assumptions, and evaluation methodology, are now explicitly included, substantially strengthening the submission
- **Value for Cooperative AI and safety**: This work speaks directly to a pressing question in Cooperative AI: *how to learn reward functions that encode system-level resilient behavior in mixed-motive environments*. Reviewers agreed that the problem is timely and valuable, and our framework provides a well-defined foundation upon which future MARL and safety research can build.

These additions further strengthen our main contribution, a principled reward-design method grounded in a cooperative resilience metric, capable of inferring a collective reward signal that drives agents toward sustained system performance under disruptions. In this way, the paper now delivers a clear conceptual advance, a rigorous and transparent technical formulation, and robust empirical validation. We hope this consolidated summary assists the AC in their meta-review and accurately reflects the improvements achieved during the rebuttal phase. It would be an honor to present these results at the conference, and we deeply appreciate your consideration.

---

### Meta-Review · Area_Chair_7yPt · 2025-12-17

**Summary:**

The reviewers highlighted significant methodological and theoretical flaws in the proposed framework. One concern was the circularity of the evaluation, as the resilience metric used to rank trajectories for training was also the main measure of success. There was also a question raised the quality of the training signal, noting that rankings were derived from random policies, potentially introducing noise rather than meaningful cooperative data. Theoretically, one reviewer argued the problem formulation was unsound, pointing out that treating a partially observable, multi-agent environment like Harvest as a fully observable MDP with shared information violates standard assumptions for mixed-motive games.

Experimentally, the reviewers found the baselines insufficient, for example, noting the absence of modern cooperative MARL algorithms such as MAPPO or HAPPO and related incentive-shaping methods. They also questioned the generalization of the results, observing that the method was tested only on small instances of a single domain and that handcrafted features outperformed learned representations, suggesting the success relied more on expert engineering than the learning algorithm itself. Finally, the lack of sensitivity analysis for the complex, manually constructed resilience metric raised doubts about the robustness of the findings.

**Reviewer Concerns:**

I believe that the authors addressed some of the concerns around the theoretical flaws, e.g. the explained why there was no circulartity issue well and they attended to the concerns regarding MDP structure, indicator scaling, observability assumptions, and metric aggregation.

However, I do not believe that they fully attended to the concerns raised regarding generalizability. As well, they did not add any additional baselines.

**Reviewer Scores:**

Given the considerations above, I would expect one or two of the reviewers would have raised their scores. (The reviewers never responsed, though, so this is a guess.)

The initial scores were 4,4,6,2 - my guess is they would have been something like 5,4,6,3 after rebuttal. That would make this a borderline paper, leaning to reject.

---

### Decision · Program_Chairs · 2026-01-26

Reject